# Preference-Guided Diffusion for Multi-Objective Offline Optimization

**Yashas Annadani**[* 1,3]    **Syrine Belakaria**[2]    **Stefano Ermon**[2]
**Stefan Bauer**[1,3]    **Barbara Engelhardt**[2,4]
[1] TU Munich    [2] Stanford University
[3] Helmholtz AI, Munich    [4] Gladstone Institutes

## Abstract

Offline multi-objective optimization aims to identify Pareto-optimal solutions given a dataset of designs and their objective values. In this work, we propose a preference-guided diffusion model that generates Pareto-optimal designs by leveraging a classifier-based guidance mechanism. Our guidance classifier is a preference model trained to predict the probability that one design dominates another, directing the diffusion model toward optimal regions of the design space. Crucially, this preference model generalizes beyond the training distribution, enabling the discovery of Pareto-optimal solutions outside the observed dataset. We introduce a novel diversity-aware preference guidance, augmenting Pareto dominance preference with diversity criteria. This ensures that generated solutions are optimal and well-distributed across the objective space, a capability absent in prior generative methods for offline multi-objective optimization. We evaluate our approach on various continuous offline multi-objective optimization tasks and find that it consistently outperforms other inverse/generative approaches while remaining competitive with forward/ surrogate-based optimization methods. Our results highlight the effectiveness of classifier-guided diffusion models in generating diverse and high-quality solutions that approximate the Pareto front well.

## 1   Introduction

Several design problems in science and engineering require optimizing a black-box, expensive-to-evaluate function. For example, in antibiotic drug discovery, the goal is to identify novel molecules with high antibacterial activity [39]. This can be formulated as a single-objective optimization problem. However, in practice, most real-world design challenges involve balancing multiple conflicting objectives. For example, in drug discovery, in addition to maximizing antibacterial activity, we also want to minimize toxicity and production costs [38]. This constitutes a multi-objective experimental design problem.

Prior work in both single and multi-objective optimization (MOO) has largely focused on adaptive experimental design using online methods such as Bayesian optimization [33]. These approaches rely on training surrogate models for each objective function and designing acquisition functions that are typically optimized via gradient-based techniques [3] or evolutionary algorithms to determine the next candidate for evaluation. This process is iteratively repeated to optimize the objectives. However, in many real-world applications, sequential evaluations—where inputs are tested one at a time or in small batches—are impractical. In some cases, we have only a single opportunity to evaluate the function, and we must allocate the entire evaluation budget efficiently.

---

[*]Correspondence to yashas.annadani@gmail.com. Code available at https://github.com/yannadani/pgd_moo.

For example, in drug design, scientists cannot test molecules one by one in wet lab experiments due to the high cost, slow turnaround, and the inherently parallelizable nature of the process [38]. Instead, it is common to evaluate all candidate molecules in a single batch. This setting is referred to as offline black-box optimization [41]. While recent work has explored offline optimization in the single-objective setting [23, 46], where extensive prior data is leveraged to model the objective function and identify potential optima, the MO case remains relatively underexplored.

Offline black-box optimization presents distinct challenges compared to traditional online optimization. Since the algorithm cannot iteratively refine the learned model using newly acquired data, it must effectively leverage the available dataset to generalize beyond observed data points. This is particularly challenging because the true optima are often expected to lie outside the existing dataset, requiring robust extrapolation. Additionally, the goal is not merely to identify data points with high function values but to find solutions that satisfy a well-defined notion of optimality, such as the *Pareto* optimality.

Prior work in single-objective offline optimization can generally be categorized into two main approaches: forward and inverse methods. Forward approaches attempt to mimic strategies used in online optimization while leveraging offline data [41]. These methods train a surrogate model of the objective function and optimize it using gradient-based techniques to propose a set of promising inputs for evaluation. They are effective when the search space is well-defined. In contrast, inverse approaches use generative models to learn an inverse mapping from function values to inputs, enabling the generation of new candidates with potentially high objective values [6, 16, 24, 25]. This distinction is critical when optimal inputs are not known in advance. For example, in chemistry, if the goal is to evaluate a known molecule, surrogate models are effective in predicting its properties. However, if the goal is to discover entirely new molecules with desired properties, inverse methods are essential, as they directly generate novel candidates rather than selecting from a predefined space. Some generative modeling approaches also draw inspiration from online methods by constructing synthetic optimization trajectories from offline data, aiming to generate new optimal points by extrapolating from the learned trajectories [23].

Multi-objective offline optimization introduces additional challenges beyond those encountered in the single-objective setting. In single-objective optimization, the goal is simply to maximize (or minimize) a function value. However, in the multi-objective case, we seek to achieve best trade-offs among competing objectives, which is formalized by Pareto optimality. Beyond identifying Pareto-optimal solutions, another critical challenge is ensuring diversity on the Pareto front. A well-structured Pareto front should provide solutions spread across different regions of the objective space, representing a broad range of Pareto-optimal designs. Although the problem of ensuring diversity of solutions is addressed in online MOO [1], ensuring both optimality and diversity is challenging in the offline setting as the algorithm must infer these solutions solely from existing data.

Xue et al. [45] has recently explored benchmarking offline multi-objective optimization (MOO) by building offline datasets for a variety of MOO benchmarks and proposing several potential algorithms. Their work extends some offline single-objective optimization (SOO) approaches to the MOO setting, such as fitting surrogate models to the offline data and optimizing over the surrogates using evolutionary algorithms. However, this work has not explored the possibility of using generative models for offline MOO.

Our contributions are:

- We formulate offline MOO as an inverse problem, and propose a novel algorithm based on diffusion models with preference-based classifier guidance. This classifier is trained to compare two candidate designs and determine which is more likely to dominate the other in the objective space.

- We address the diversity challenge in the offline MOO setting. Rather than just identifying high-value designs, we train the preference model to favor solutions that are not only optimal but also well-distributed across the objective space.

- We demonstrate through extensive experiments on both synthetic and real-world settings that our approach performs better than other inverse methods in terms of convergence to the Pareto front and the diversity of obtained solutions.

- Among inverse techniques, our approach consistently achieves the strongest results; making it a competitive choice in cases where forward methods are not feasible. Even when forward

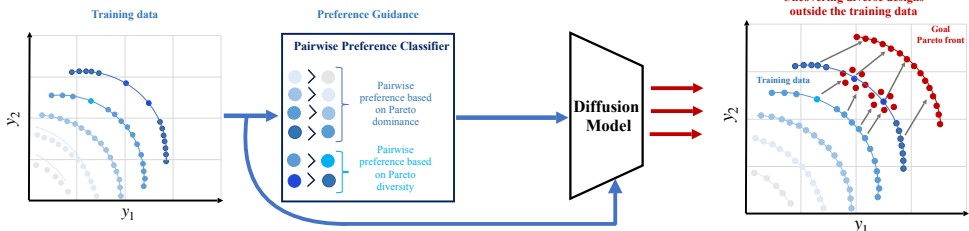

Figure 1: A schematic representation of the proposed preference guided diffusion approach for offline multi-objective optimization.

methods are applicable, our algorithm remains competitive while maintaining the benefits of generative exploration.

## 2 Background

### 2.1 Offline Multi-Objective Optimization

Multi-objective optimization (MOO) seeks to find the design $\boldsymbol{x} \in \mathcal{X}^d$ that minimizes (or maximizes) a set of $m$ different objectives:

$$\min_{\boldsymbol{x} \in \mathcal{X}^d} \boldsymbol{f}(\boldsymbol{x}) \coloneqq \{f_1(\boldsymbol{x}), \ldots, f_m(\boldsymbol{x})\}, \tag{1}$$

where $f_i : \mathcal{X}^d \mapsto \mathbb{R}$ is an unknown and expensive-to-evaluate objective. For most practical problems, the objectives are not simultaneously optimizable by a single design. Hence, the goal instead is to find the set of designs that are *Pareto optimal*.

**Definition 2.1** (Pareto Dominance). A design $\boldsymbol{x}$ *Pareto dominates* another design $\boldsymbol{z} \in \mathcal{X}^d$ (denoted by $\boldsymbol{x} \prec \boldsymbol{z}$) if $f_i(\boldsymbol{x}) \leq f_i(\boldsymbol{z}) \; \forall i$ and $\exists j : f_j(\boldsymbol{x}) < f_j(\boldsymbol{z})$.

**Definition 2.2** (Pareto Optimality and Pareto Front). A design $\boldsymbol{x}^*$ is *Pareto optimal* if $\nexists \boldsymbol{x} \in \mathcal{X}^d$ such that $\boldsymbol{x} \prec \boldsymbol{x}^*$. The set of all Pareto optimal designs is called a *Pareto set*. Correspondingly, the objective values of the Pareto set $\{\boldsymbol{f}(\boldsymbol{x}^*) \mid \boldsymbol{x}^*\}$ is called the *Pareto front (PF)*.

The Pareto front provides an optimal set of trade-offs that can be achieved from the objectives when they are not simultaneously optimizable.

Sequential methods that collect data by selecting designs and evaluating their function values are the most common approach for MOO, making use of surrogate models with uncertainty quantification to learn the unknown objectives. However, for many practical problems, these sequential methods are not feasible due to prohibitive cost or time constraints (or both). Instead of iteratively allocating an evaluation budget to refine the design choices, offline optimization uses the entire budget in a single round of function evaluations. In offline optimization, we have access to a dataset of $N$ non-optimal design-objective values pairs $\mathcal{D} \coloneqq \{(\boldsymbol{x}^{(i)}, \boldsymbol{f}(\boldsymbol{x}^{(i)})\}_{i=1}^N$. The goal of offline MOO is to find the Pareto-optimal set by relying only on the existing dataset $\mathcal{D}$.

### 2.2 Diffusion Models

Diffusion models [17, 34, 35] are a class of generative models defined by a Markov chain that sequentially adds noise to data samples and then learns to denoise from white noise (i.e., generate a sample) by reversing the Markov chain. In this work, we follow the Denoising Diffusion Probabilistic Models (DDPM) [17] approach, which we summarize here. Given a sample $\boldsymbol{x}_0 \sim q(\boldsymbol{x})$, a time-dependent forward noising process is defined as:

$$q(\boldsymbol{x}_t \mid \boldsymbol{x}_{t-1}) \coloneqq \mathcal{N}(\boldsymbol{x}_t; \sqrt{1 - \beta_t} \boldsymbol{x}_{t-1}, \beta_t \mathbb{I}_d), \tag{2}$$

where $\beta_t$ is the variance of the noising schedule at timestep $t$ such that $\beta_1 < \beta_2 < \cdots < \beta_T$, and $T$ is the total number of timesteps. Let $\alpha_t = 1 - \beta_t$ and $\tilde{\alpha}_t = \prod_{i=1}^t \alpha_t$; then the noised sample $\boldsymbol{x}_t$ can

be obtained in closed form:

$$\boldsymbol{x}_t = \sqrt{\tilde{\alpha}_t}\boldsymbol{x}_0 + \sqrt{1 - \tilde{\alpha}_t}\epsilon_t, \quad \epsilon_t \sim \mathcal{N}(0, \mathbb{I}_d). \tag{3}$$

The reverse process conditional $q(\boldsymbol{x}_{t-1} \mid \boldsymbol{x}_t)$ is not tractable. Therefore, a denoising model defined as $p_\theta(\boldsymbol{x}_{t-1} \mid \boldsymbol{x}_t) \coloneqq \mathcal{N}(\boldsymbol{x}_{t-1}; \mu_\theta(\boldsymbol{x}_t), \Sigma_\theta(\boldsymbol{x}_t))$ is learned by optimizing parameters $\theta$. Instead of parameterizing the reverse process to estimate $\mu_\theta(\boldsymbol{x}_t)$, it is common to reparameterize this reverse process to predict the noise that was added to produce $\boldsymbol{x}_t$ (equation 3). If $\epsilon_\theta(\boldsymbol{x}_t, t)$ is the denoising model to predict the added noise, reparameterization yields the mean $\mu_\theta(\boldsymbol{x}_t)$ as:

$$\mu_\theta(\boldsymbol{x}_t) = \frac{1}{\sqrt{\alpha_t}}\left(\boldsymbol{x}_t - \frac{1-\alpha_t}{\sqrt{1-\tilde{\alpha}_t}}\epsilon_\theta(\boldsymbol{x}_t, t)\right). \tag{4}$$

The reverse process variance is set to be the same as the forward process variance at time $t$, i.e., $\Sigma_\theta(\boldsymbol{x}_t) \coloneqq \beta_t \mathbb{I}_d$. The denoising model can be trained with mean squared error (MSE) loss:

$$\ell(\theta) = \mathop{\mathbb{E}}_{t, \epsilon_t, \boldsymbol{x}_0}\left[\|\epsilon_t - \epsilon_\theta(\boldsymbol{x}_t, t)\|_2^2\right]. \tag{5}$$

A new sample can be generated by first sampling $\tilde{\boldsymbol{x}}_T \sim \mathcal{N}(0, \mathbb{I}_d)$ and then autoregressively sampling from $\mathcal{N}(\boldsymbol{x}_{t-1}; \mu_\theta(\tilde{\boldsymbol{x}}_t), \beta_t \mathbb{I}_d)$ to get $\tilde{\boldsymbol{x}}_0$.

### 2.3  Classifier Guidance

If label $\boldsymbol{o}$ corresponding to each sample $\boldsymbol{x}_0$ is available, then *classifier guidance* allows one to generate new samples from a trained diffusion model specific to a desired label. To do this, classifier guidance [13] trains an additional time-dependent classifier of the input $p_\phi(\boldsymbol{o} \mid \boldsymbol{x}_t, t)$. Along with the trained denoising model $\epsilon_\theta(\boldsymbol{x}_t, t)$, a new conditional sample can be generated by first sampling $\tilde{\boldsymbol{x}}_T \sim \mathcal{N}(0, \mathbb{I}_d)$ and then, from $t = T$ to $t = 1$, autoregressively sampling:

$$\tilde{\boldsymbol{x}}_{t-1} \mid \boldsymbol{o}, \tilde{\boldsymbol{x}}_t \sim \mathcal{N}(\boldsymbol{x}_{t-1}; \mu_\theta(\tilde{\boldsymbol{x}}_t) + w\beta_t \nabla_{\tilde{\boldsymbol{x}}_t} \log p_\phi(\boldsymbol{o} \mid \tilde{\boldsymbol{x}}_t, t), \beta_t \mathbb{I}_d), \tag{6}$$

where $w$ is the *guidance strength*. In this work, we use classifier guidance to generate samples that approximate the optimal Pareto sets where the classifier is a preference model that predicts the dominance of one input over the other.

## 3  Related Work

**Online Multi-Objective Black-Box Optimization:**    Adaptive experimental design has primarily explored multi-objective optimization (MOO) in an online or sequential fashion, where solutions are iteratively refined as new data arrives [1, 4]. Although less studied than single-objective optimization, several sequential approaches have proven effective. The most established is Bayesian optimization (BO) [18, 33], which typically uses Gaussian processes [44] to model objectives, especially in data-scarce regimes. Multi-objective extensions of BO often reduce the problem via scalarization [21] or employ acquisition functions such as expected hypervolume improvement (EHVI) [15] and information gain [4] to balance trade-offs across objectives. Recent work also introduced batch selection strategies [11], though most methods emphasize Pareto dominance over diversity. A few exceptions, such as Konakovic Lukovic et al. [22] and Ahmadianshalchi et al. [1], explicitly promote Pareto front diversity. Beyond traditional BO, neural approaches extend online MOO using generative models like variational autoencoders (VAEs) combined with Gaussian processes over the latent space [36]. However, these methods inherit VAE limitations—posterior collapse and non-identifiability—that restrict their robustness in practical optimization tasks.

**Offline Single-Objective Black-Box Optimization:**    Offline single-objective optimization focuses on leveraging existing datasets without additional data collection. Forward methods [41, 46] train surrogate models to approximate the objective function and then optimize these surrogates for high-performing inputs. Inverse methods instead employ conditional GANs [6, 16, 25] to learn mappings from function outputs back to inputs. To bridge forward and inverse strategies, Chemingui et al. [7], Krishnamoorthy et al. [23] proposed hybrid methods that synthesize pseudo-optimization trajectories from offline data, mimicking online behavior. Diffusion-based frameworks [24] adjust diffusion losses with weighted importance terms, while classifier-guided models [8] bias generation toward globally optimal designs. Kim et al. [19] provides a comprehensive review of model-based offline optimization methods, summarizing progress across forward, inverse, and hybrid approaches. Additionally, Trabucco et al. [42] established a standardized benchmarking suite for consistent evaluation of offline optimization algorithms.

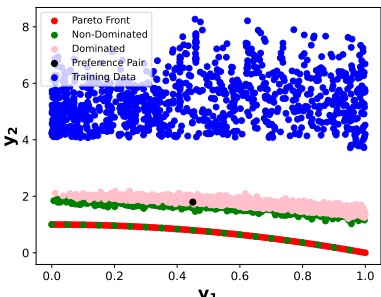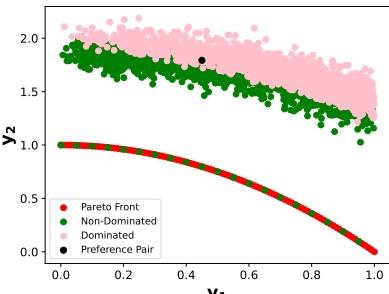

Figure 2: Generalization of the preference model on regions unseen in the training data on the ZDT2 task [53]. The preference model gives good prediction of Pareto dominance between the reference design (in black) with other designs (in pink and green). Pink indicates that the preference model predicts these designs to be dominated by the reference design and green indicates that these designs are predicted as dominating the reference design. The figure on the right is a zoomed-in version of the left, excluding the training data (in blue).

**Offline Multi-Objective Black-Box Optimization:** Research on offline multi-objective optimization (MOO) remains limited. Xue et al. [45] recently proposed a benchmarking framework with offline datasets and baseline algorithms across MOO benchmarks. Their work mainly extends forward-style single-objective methods by fitting surrogate models to offline data and optimizing them via evolutionary algorithms but does not explore generative or inverse techniques. Concurrently, Yuan et al. [48] introduced ParetoFlow, a flow-based generative model for offline MOO that embeds objective weighting directly into the loss function, effectively scalarizing multiple objectives. While this inverse approach is promising, it does not explicitly address Pareto front diversity—an essential factor for capturing representative trade-offs in MOO.

## 4    Preference-Guided Diffusion for Offline Multi-Objective Optimization

We present a new effective approach for offline MOO by using classifier guidance to generate samples from Pareto optimal sets with a diffusion model trained on offline data. Our approach does not require training individual surrogate models for each objective. It relies on an inverse strategy while ensuring the ability to generate diverse samples from the Pareto optimal set. We refer to our method as *preference-guided diffusion for multi-objective offline optimization* (PGD-MOO).

Let $\boldsymbol{x} \in \mathcal{X}^d \subseteq \mathbb{R}^d$ be any $d$-dimensional design with corresponding objective values $y_i = f_i(\boldsymbol{x})$ defined by unknown and expensive-to-evaluate functions $f_i : \mathcal{X}^d \mapsto \mathbb{R}$. Let $\boldsymbol{y} = [y_1, \ldots, y_m]^T$ be the vector of objective values for an $m$-objective problem. In offline MOO, we have access to a dataset $\mathcal{D} := \{\boldsymbol{x}^{(i)}, \boldsymbol{y}^{(i)}\}_{i=1}^N$ of $N$ previously evaluated design-objective pairs. Given $\mathcal{D}$, the goal is to generate designs $\boldsymbol{x}^*$ from the unknown optimal Pareto set.

While diffusion models capture the distribution over data $p(\boldsymbol{x})$, in offline MOO, we are often interested in samples that lie outside the training data, closer to the Pareto front. This motivates the use of classifier guidance. Directly using classifier guidance in diffusion models involves training surrogate models for each objective, which often requires scalarization and hence can be suboptimal. We propose to use preference-based guidance to capture Pareto dominance relations between data points.

### 4.1    Preference Guided Diffusion

In this work, we explore an alternate guidance strategy that does not involve training surrogate models for every objective. Instead, we model the space of multiple-objectives through preference pairs, as defined by a Bradley-Terry model [5] of designs.

In this regard, we train a preference model that predicts the (log)- probability of whether a design Pareto dominates (Definition 2.1) another design. During sampling of the designs, Given two designs $\boldsymbol{x}$ and $\hat{\boldsymbol{x}}$, we train a (time-conditioned) binary classifier that predicts $\log p_\phi(\boldsymbol{x} \prec \hat{\boldsymbol{x}} \mid \boldsymbol{x}, \hat{\boldsymbol{x}}, t)$. We parameterize this distribution with a multi-layer perceptron (MLP) that takes in two inputs (designs) of size $2 \times d$ and outputs the logit of the Bernoulli distribution predicting whether the first input Pareto dominates the second input.

---

**Algorithm 1** Sampling from Preference Guided Diffusion

---

**Require:** Trained $\epsilon_\theta(\boldsymbol{x}_t, t)$, preference model $p_\phi(\boldsymbol{x} \prec \hat{\boldsymbol{x}} \mid \boldsymbol{x}, \hat{\boldsymbol{x}}, t)$, guidance weight $w$ and the most dominant design in the dataset $\boldsymbol{x}^{\mathcal{D}(\text{best})}$

1: $r \leftarrow \boldsymbol{x}^{\mathcal{D}(\text{best})}$
2: $\tilde{\boldsymbol{x}}_T \sim \mathcal{N}(0, \mathbb{I}_d)$
3: **for** t = T to 1 **do**
4:     $\mu_\theta(\tilde{\boldsymbol{x}}_t) = \frac{1}{\sqrt{\alpha_t}} \left( \tilde{\boldsymbol{x}}_t - \frac{1-\alpha_t}{\sqrt{1-\bar{\alpha}_t}} \epsilon_\theta(\tilde{\boldsymbol{x}}_t, t) \right)$
5:     Compute preference score $s_p = \nabla_{\tilde{\boldsymbol{x}}_t} p_\phi(\tilde{\boldsymbol{x}}_t \prec r \mid \tilde{\boldsymbol{x}}_t, r, t)$
6:     Sample $\tilde{\boldsymbol{x}}_{t-1} \sim \mathcal{N}(\boldsymbol{x}_{t-1}; \mu_\theta(\tilde{\boldsymbol{x}}_t) + w\beta_t s_p, \beta_t \mathbb{I}_d)$
7:     $r \leftarrow \tilde{\boldsymbol{x}}_t$
8: **end for**
9: **return** $\tilde{\boldsymbol{x}}_0$

---

**Training the Preference Classifier.** To train the preference classifier, we first sort the points in the training data by their Pareto dominance. This divides the data into multiple fronts in increasing order of dominance, with points in the same front considered equally dominant. Next, we select $(\boldsymbol{x}, \hat{\boldsymbol{x}})$ pairs randomly from the dataset. If one design strictly dominates the other, it is labeled as preferred. If both $\boldsymbol{x}$ and $\hat{\boldsymbol{x}}$ are equally dominant, we assign the label $\boldsymbol{x} \prec \hat{\boldsymbol{x}}$ if $\boldsymbol{x}$ has more *diversity contribution* than $\hat{\boldsymbol{x}}$ wrt other points that belong to the front. The diversity contribution of each point is calculated using the crowding distance [12] of a selected design w.r.t all other points that belong to the same front in the dataset. Crowding distance for any point $\boldsymbol{x}$ is computed as follows:

$$d_{\text{CD}}(\boldsymbol{x}) = \sum_{i=1}^{m} \frac{y_i^+ - y_i^-}{y_i^{\max} - y_i^{\min}}, \tag{7}$$

where $y_i^+$ and $y_i^-$ are the $i^{\text{th}}$ objective values of neighboring designs of $\boldsymbol{x}$ in the corresponding front sorted according to $i$th objective value. $y_i^{\max}$ and $y_i^{\min}$ are the maximum and minimum values of the objective $i$ in the current front. Crowding distance has been used as a secondary selection criterion in evolutionary algorithms such as NSGA-2 [12] to maintain the diversity of solutions. Using crowding distance to create a binary label encourages the preference model to not only guide the diffusion model towards more Pareto-dominant regions, but also ensure that the designs that make up the Pareto front are diverse. For the denoising model, we train an unconditional diffusion model $\epsilon_\theta(\boldsymbol{x}_t, t)$ (§2.2) on the designs $\boldsymbol{x}$ in the dataset, similar to DDPM [17].

**Sampling Designs.** With a trained denoising model $\epsilon_\theta(\boldsymbol{x}_t, t)$ and preference model $p_\phi(\boldsymbol{x} \prec \hat{\boldsymbol{x}} \mid \boldsymbol{x}, \hat{\boldsymbol{x}}, t)$, we sample a new design by using classifier guidance (§2.3). We input both the denoised variable at the current timestep $\tilde{\boldsymbol{x}}_t$ as well as from the previous realization of denoising, i.e., $\tilde{\boldsymbol{x}}_{t+1}$ for the preference model to estimate $\nabla_{\tilde{\boldsymbol{x}}_t} p_\phi(\tilde{\boldsymbol{x}}_t \prec \tilde{\boldsymbol{x}}_{t+1} \mid \tilde{\boldsymbol{x}}_t, \tilde{\boldsymbol{x}}_{t+1}, t)$. At the beginning of the denoising procedure when $\boldsymbol{x}_{t+1}$ is not defined, we input the most dominant design in the dataset for preference comparison. The most dominant design in the dataset is chosen based on non-dominated sorting. The sampling procedure is summarized in Algorithm 1. Intuitively, a preference model that generalizes well beyond its training data should guide the denoising process such that, at each step of denoising, the resulting sample $\tilde{\boldsymbol{x}}_t$ is in a more Pareto-dominant region.

With enough timesteps, the resulting denoised sample $\tilde{\boldsymbol{x}}_0$ will be close to the Pareto front. Moreover, the connection to Bradley-Terry model of preference is apparent: the gradient of the logit which indicates the probability of the first input dominating the second, carries information about the reward [32]. Here, we use this to guide the diffusion model such that it can sample from regions closer to the Pareto front through the denoising process.

This approach does not require training surrogate models, thus providing a simple alternative approach to offline MOO. We find that the preference model generalizes well outside of the training data (see Fig. 2), therefore providing guidance to the diffusion model to generate designs outside of the training data close to the Pareto front, while maintaining diversity in the samples.

# 5 Experiments

We perform several experiments on standard benchmarks for offline MOO. Through these experiments, we would like to understand how close the generated samples are to the Pareto front, as well as the diversity of the solutions.

**Benchmark Tasks:** Our evaluation closely follows the benchmarking effort provided in prior work [45]. We evaluate our approach on two sets of tasks: **synthetic** and real-world applications-based **RE** engineering suite [40]. Each task consists of a dataset of 60k offline datapoints. As in [45], we use 54k randomly chosen data points for training and the remaining for validation.

**1. Synthetic** set of tasks consist of 12 distinct tasks each with their own dataset. The objectives are analytic functions with tractable Pareto-fronts. This set of tasks has been widely used in MOO problems. Every task consists of 2-3 objectives with $d$ ranging from 10 to 30.

**2. RE** engineering suite of problems are set of 12 distinct tasks, each with their own dataset. These tasks are based on real-world applications in engineering, for instance, rocket injector design and disc brake design. $d$ ranges from 3-7 variables and the number of objectives $m$ varies from 2 to 6.

**3. MO-NAS** are set of tasks are based on neural architecture search benchmarks from [14, 27, 30] which consists of multiple objectives like prediction error and hardware constraints like GPU latency. In particular, we use the datasets corresponding to C10/MOP(1-9) and IN1K/MOP(1-9) tasks provided by Lu et al. [30]. As the design space is discrete, we operate on the continuous space of corresponding logits, as in prior work [45, 48].

Details of individual tasks and their corresponding datasets are given in Appendix B.

**Baselines:** We compare our approach with two categories of baselines:

**1.** We compare with **ParetoFlow** [48], a classifier guided generative model based on flow-matching [28]. Classifiers for guidance are trained surrogate models for each objective, followed by scalarization. ParetoFlow is our **primary baseline** since our work is mainly concerned with studying inverse approaches for offline MOO.

**2.** Forward approaches using evolutionary algorithms: As suggested in prior work [45], a standard approach to offline MOO is to train a surrogate

Table 1: Average ranking of hypervolume obtained by different methods across synthetic, RE and MO-NAS tasks.

| Method | Synthetic | RE | MO-NAS |
|---|---|---|---|
| MultiHead | 7.5 | 5.73 | 10.11 |
| MultiHead - PcGrad | 6.08 | 6.06 | 8.00 |
| MultiHead - GradNorm | 9.75 | 11.4 | 8.50 |
| MultipleModels | 6.5 | **3.67** | 9.39 |
| MultipleModels - COM | 7.83 | 7.27 | 2.72 |
| MultipleModels - IOM | 5.58 | 4.6 | 5.44 |
| MultipleModels - ICT | 6.67 | 4.67 | 6.61 |
| MultipleModels - RoMA | 6.58 | 7.67 | 7.44 |
| MultipleModels - TriMentoring | 8.33 | 4.3 | 8.11 |
| ParetoFlow | 7.58 | 6.17 | 4.67 |
| **PGD-MOO + Data Pruning (Ours)** | 5.08 | 9.06 | 2.14 |
| **PGD-MOO (Ours)** | **4.42** | 7.67 | **1.86** |

model for each objective and then use evolutionary algorithms such as NSGA-2 [12] to search over the design space. Although there are various ways to learn surrogate models, we compare with deep neural network (DNN)-based approaches, which are shown to perform best according to benchmarks [45]. The DNN approaches we compare with are: i) **A Multi-Head** Model: Uses multi-task learning [50] to train a joint surrogate for all objectives. Training techniques for this approach such as GradNorm [10] and PcGrad [47] are also compared. ii) **Multiple Models**: Maintain $m$ independent surrogate models, each making use of a single optimization technique, including COMs [41], ROMA [46], IOM [31], ICT [49], and Tri-mentoring [9].

Other forward approaches like multi-objective Bayesian optimization algorithms are shown to perform worse than evolutionary algorithms on both synthetic and RE set of tasks [45], hence we exclude them from comparisons in this work.

**Evaluation Metrics:** For each algorithm, we evaluate the convergence of solutions using the hypervolume metric [52], a standard metric in MOO for measuring the closeness of the proposed designs to the Pareto front. *Hypervolume* measures the volume of the objective space between a

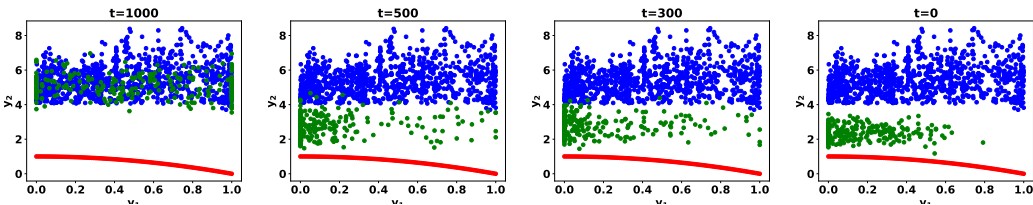

Figure 3: Plot of the samples from our preference-guided diffusion model (in green) on the ZDT2 task [53] at different timesteps of denoising. Convergence of samples close to the Pareto front (in red) outside of the training data (blue) highlights the importance of preference guidance.

Table 2: Hypervolume results of DTLZ subtasks (part of the **synthetic** task). Each method is run for five random seeds and evaluated on 256 designs.

| Method | DTLZ1 | DTLZ2 | DTLZ3 | DTLZ4 | DTLZ5 | DTLZ6 | DTLZ7 |
|---|---|---|---|---|---|---|---|
| D (best) | 10.60 | 9.91 | 10.00 | 10.76 | 9.35 | 8.88 | 8.56 |
| MultiHead | $10.51 \pm 0.23$ | $9.03 \pm 0.56$ | $10.48 \pm 0.23$ | $6.73 \pm 1.4$ | $8.41 \pm 0.15$ | $8.72 \pm 1.07$ | $\mathbf{10.66 \pm 0.09}$ |
| MultiHead - PcGrad | $10.64 \pm 0.01$ | $9.64 \pm 0.33$ | $10.55 \pm 0.12$ | $9.95 \pm 1.93$ | $9.02 \pm 0.24$ | $9.90 \pm 0.25$ | $10.61 \pm 0.03$ |
| MultiHead - GradNorm | $10.64 \pm .01$ | $8.86 \pm 1.27$ | $10.26 \pm 0.28$ | $7.45 \pm 0.75$ | $7.87 \pm 1.06$ | $8.16 \pm 2.21$ | $10.31 \pm 0.22$ |
| MultipleModels | $10.64 \pm 0.01$ | $9.03 \pm 0.80$ | $10.58 \pm 0.03$ | $7.66 \pm 1.3$ | $7.65 \pm 1.39$ | $9.58 \pm 0.31$ | $10.61 \pm 0.16$ |
| MultipleModels - COM | $10.64 \pm 0.01$ | $8.99 \pm 0.97$ | $10.27 \pm 0.37$ | $9.72 \pm 0.39$ | $9.44 \pm 0.41$ | $9.37 \pm 0.35$ | $10.09 \pm 0.36$ |
| MultipleModels - IOM | $10.64 \pm 0.01$ | $10.10 \pm 0.27$ | $10.24 \pm 0.13$ | $10.03 \pm 0.53$ | $9.77 \pm 0.18$ | $9.30 \pm 0.31$ | $10.60 \pm 0.05$ |
| MultipleModels - ICT | $10.64 \pm 0.01$ | $8.68 \pm 0.88$ | $10.25 \pm 0.42$ | $10.33 \pm 0.24$ | $9.25 \pm 0.28$ | $9.10 \pm 1.16$ | $10.29 \pm 0.05$ |
| MultipleModels - RoMA | $10.64 \pm 0.01$ | $10.04 \pm 0.05$ | $10.61 \pm 0.03$ | $9.25 \pm 0.11$ | $8.71 \pm 0.47$ | $9.84 \pm 0.25$ | $10.53 \pm 0.04$ |
| MultipleModels - TriMentoring | $10.64 \pm 0.01$ | $9.39 \pm 0.35$ | $10.48 \pm 0.12$ | $10.21 \pm 0.06$ | $7.69 \pm 1.03$ | $9.00 \pm 0.48$ | $10.12 \pm 0.09$ |
| ParetoFlow | $10.60 \pm 0.02$ | $10.13 \pm 0.16$ | $10.41 \pm 0.09$ | $10.29 \pm 0.17$ | $9.65 \pm 0.23$ | $9.25 \pm 0.43$ | $8.94 \pm 0.18$ |
| **PGD-MOO + Data Pruning (Ours)** | $10.64 \pm 0.01$ | $\mathbf{10.55 \pm 0.01}$ | $\mathbf{10.63 \pm 0.01}$ | $\mathbf{10.63 \pm 0.01}$ | $\mathbf{10.07 \pm 0.02}$ | $\mathbf{10.15 \pm 0.03}$ | $9.57 \pm 0.07$ |
| **PGD-MOO (Ours)** | $\mathbf{10.65 \pm 0.01}$ | $\mathbf{10.55 \pm 0.01}$ | $\mathbf{10.63 \pm 0.01}$ | $\mathbf{10.64 \pm 0.01}$ | $10.06 \pm 0.02$ | $10.14 \pm 0.01$ | $9.70 \pm 0.18$ |

reference point and the objective vectors of the solution set, and does not require access to the true Pareto front. The reference point used for evaluation of hypervolume is taken from [45](Appendix B).

In addition to the hypervolume, we also measure the diversity of the obtained solutions using the $\Delta$-spread metric [12, 51]. The $\Delta$-spread measures the extent of the spread achieved in a computed Pareto front approximation [2]. It is important to consider the diversity of the obtained solutions, especially in the case of MOO wherein there is no single "best" design, but rather an entire set of solutions based on the Pareto front. In addition, in the case of offline optimization, the acquisition is single-shot. Therefore, solutions that are diverse and hence provide more coverage over the objective space are preferable. In this work, we provide the first effort to evaluate and benchmark the diversity of solutions obtained by different approaches in offline MOO. Further details of the evaluation metrics and their computation is provided in Appendix A.1.

We evaluate all methods on 5 random seeds, and compute the metrics using a budget of 256 designs.

## 5.1 Training Details

We parameterize the unconditional denoising model to be a multi-layer perceptron (MLP) with two 512-dimensional hidden layers, followed by a ReLU nonlinearity and layer normalization [26]. We also incorporate sinusoidal time embedding [43] for conditioning. We parameterize the preference model to be an MLP with three hidden layers, with first two hidden layers having the same number of units as the input, while the last hidden layer is having 512 units. Similar to denoising model, we also use ReLU nonlinearity followed by layer normalization and sinusoidal time embedding.

The denoising model is trained with AdamW optimizer [29] with learning rate of $5e-4$ for up to 200 epochs. Following Ho et al. [17], we employ a linear noise schedule such that the noise $\beta_t$ grows linearly from $1e-4$ to $0.02$. The preference model is trained with Adam optimizer [20] with learning rate of $1e-5$ for up to 500 epochs. During sampling, we set the guidance weight $w$ to 10. For the preference model, we also experiment with pruning the training data to only contain the top $30\%$ of points, sorted according to their dominance. We refer to this method as **PGD-MOO + Data Pruning** in the results.

Table 3: Hypervolume results of ZDT subtasks (part of the **synthetic** task). Each method is run for five random seeds and evaluated on 256 designs.

| Method | ZDT1 | ZDT2 | ZDT3 | ZDT4 | ZDT6 |
|---|---|---|---|---|---|
| D (best) | 4.17 | 4.67 | 5.15 | 5.45 | 4.61 |
| MultiHead | $4.8 \pm 0.03$ | $5.57 \pm 0.07$ | $5.58 \pm 0.2$ | $4.59 \pm 0.26$ | $4.78 \pm 0.01$ |
| MultiHead - PcGrad | $\mathbf{4.84 \pm 0.01}$ | $5.55 \pm 0.11$ | $5.51 \pm 0.03$ | $3.68 \pm 0.70$ | $4.67 \pm 0.1$ |
| MultiHead - GradNorm | $4.63 \pm 0.15$ | $5.37 \pm 0.17$ | $5.54 \pm 0.2$ | $3.28 \pm 0.9$ | $3.81 \pm 1.2$ |
| MultipleModels | $4.81 \pm 0.02$ | $5.57 \pm 0.07$ | $5.48 \pm 0.21$ | $5.03 \pm 0.19$ | $4.78 \pm 0.01$ |
| MultipleModels - COM | $4.52 \pm 0.02$ | $4.99 \pm 0.12$ | $5.49 \pm 0.07$ | $\mathbf{5.10 \pm 0.08}$ | $4.41 \pm 0.21$ |
| MultipleModels - IOM | $4.68 \pm 0.12$ | $5.45 \pm 0.11$ | $5.61 \pm 0.06$ | $4.99 \pm 0.21$ | $4.75 \pm 0.01$ |
| MultipleModels - ICT | $4.82 \pm 0.01$ | $5.58 \pm 0.01$ | $5.59 \pm 0.06$ | $4.63 \pm 0.43$ | $4.75 \pm 0.01$ |
| MultipleModels - RoMA | $\mathbf{4.84 \pm 0.01}$ | $5.43 \pm 0.35$ | $\mathbf{5.89 \pm 0.04}$ | $4.13 \pm 0.11$ | $1.71 \pm 0.10$ |
| MultipleModels - TriMentoring | $4.64 \pm 0.10$ | $5.22 \pm 0.11$ | $5.16 \pm 0.04$ | $\mathbf{5.12 \pm 0.12}$ | $2.61 \pm 0.01$ |
| ParetoFlow | $4.23 \pm 0.04$ | $\mathbf{5.65 \pm 0.11}$ | $5.29 \pm 0.14$ | $5.00 \pm 0.22$ | $4.48 \pm 0.11$ |
| **PGD-MOO + Data Pruning (Ours)** | $4.54 \pm 0.08$ | $5.21 \pm 0.06$ | $5.61 \pm 0.06$ | $5.06 \pm 0.07$ | $4.56 \pm 0.14$ |
| **PGD-MOO (Ours)** | $4.41 \pm 0.08$ | $5.33 \pm 0.05$ | $5.54 \pm 0.10$ | $5.02 \pm 0.03$ | $\mathbf{4.82 \pm 0.01}$ |

Table 4: Selected results of hypervolume on **RE** task. Results are evaluated on 256 designs with five different random seeds.

| Method | RE21 | RE22 | RE25 | RE32 | RE35 | RE37 | RE41 | RE61 |
|---|---|---|---|---|---|---|---|---|
| D(best) | 4.1 | 4.78 | 4.79 | 10.56 | 10.08 | 5.57 | 18.27 | 97.49 |
| MultiHead-GradNorm | $4.28 \pm 0.39$ | $4.7 \pm 0.44$ | $4.52 \pm 0.5$ | $10.54 \pm 0.15$ | $9.76 \pm 1.3$ | $5.67 \pm 1.41$ | $17.06 \pm 3.82$ | $108.01 \pm 1.0$ |
| MultiHead-PcGrad | $4.59 \pm 0.01$ | $4.73 \pm 0.36$ | $4.78 \pm 0.14$ | $10.63 \pm 0.01$ | $10.51 \pm 0.05$ | $6.68 \pm 0.06$ | $20.66 \pm 0.1$ | $108.54 \pm 0.23$ |
| MultiHead | $4.6 \pm 0.0$ | $\mathbf{4.84 \pm 0.0}$ | $4.74 \pm 0.2$ | $10.6 \pm 0.05$ | $10.49 \pm 0.07$ | $6.67 \pm 0.05$ | $20.62 \pm 0.11$ | $108.92 \pm 0.22$ |
| MultipleModels-COM | $4.38 \pm 0.09$ | $\mathbf{4.84 \pm 0.0}$ | $4.83 \pm 0.01$ | $10.64 \pm 0.01$ | $10.55 \pm 0.02$ | $6.35 \pm 0.1$ | $20.37 \pm 0.06$ | $107.99 \pm 0.48$ |
| MultipleModels-ICT | $4.6 \pm 0.0$ | $\mathbf{4.84 \pm 0.0}$ | $\mathbf{4.84 \pm 0.0}$ | $10.64 \pm 0.0$ | $10.5 \pm 0.01$ | $6.73 \pm 0.0$ | $20.58 \pm 0.04$ | $108.68 \pm 0.27$ |
| MultipleModels-IOM | $\mathbf{4.58 \pm 0.02}$ | $\mathbf{4.84 \pm 0.0}$ | $4.83 \pm 0.01$ | $\mathbf{10.65 \pm 0.0}$ | $10.57 \pm 0.01$ | $6.71 \pm 0.02$ | $20.66 \pm 0.05$ | $107.71 \pm 0.5$ |
| MultipleModels-RoMA | $4.57 \pm 0.0$ | $4.61 \pm 0.51$ | $4.83 \pm 0.01$ | $10.64 \pm 0.0$ | $10.53 \pm 0.03$ | $6.67 \pm 0.02$ | $20.39 \pm 0.09$ | $108.47 \pm 0.28$ |
| MultipleModels-TriMentoring | $4.6 \pm 0.0$ | $4.84 \pm 0.0$ | $\mathbf{4.84 \pm 0.0}$ | $10.62 \pm 0.01$ | $\mathbf{10.59 \pm 0.0}$ | $\mathbf{6.73 \pm 0.01}$ | $20.68 \pm 0.04$ | $108.61 \pm 0.29$ |
| MultipleModels | $4.6 \pm 0.0$ | $4.84 \pm 0.0$ | $4.63 \pm 0.25$ | $10.62 \pm 0.02$ | $10.55 \pm 0.01$ | $6.73 \pm 0.03$ | $\mathbf{20.77 \pm 0.08}$ | $\mathbf{108.96 \pm 0.06}$ |
| ParetoFlow | $4.2 \pm 0.17$ | $4.86 \pm 0.01$ | - | $10.61 \pm 0.0$ | $11.12 \pm 0.02$ | $6.55 \pm 0.59$ | $19.41 \pm 0.92$ | $107.1 \pm 6.96$ |
| **PGD-MOO + Data Pruning (Ours)** | $4.42 \pm 0.04$ | $4.83 \pm 0.01$ | $\mathbf{4.84 \pm 0.0}$ | $10.64 \pm 0.0$ | $10.43 \pm 0.04$ | $5.99 \pm 0.18$ | $19.37 \pm 0.15$ | $103.04 \pm 1.71$ |
| **PGD-MOO (Ours)** | $4.46 \pm 0.03$ | $\mathbf{4.84 \pm 0.0}$ | $\mathbf{4.84 \pm 0.0}$ | $\mathbf{10.65 \pm 0.0}$ | $10.32 \pm 0.1$ | $6.13 \pm 0.12$ | $19.31 \pm 0.46$ | $105.02 \pm 1.14$ |

## 5.2 Results

**Evaluation of Convergence.** We provide detailed results of hypervolume for various baselines and our approach on the synthetic task (Tables 2 and 3). We find that our approach performs competitively with respect to baselines. Preference-guided diffusion performs on average better than ParetoFlow, another generative model-based approach using guidance. This shows the benefits of having a preference model as a classifier for guidance. Vizualization of the final sampled designs from our approach is provided in Appendix E.2. Overall, our method performs better than other baselines, which learn surrogate models and use evolutionary algorithms in the **synthetic** task setting (Fig. 3). In addition, we also find that our method performs competitively in the **RE** engineering suite (Tables 4 and 22) and the MO-NAS tasks (Table 1). In problems with higher number of objectives, we find that our approach is slightly worse compared to the baselines in terms of hypervolume. However, we note that our approach is much simpler to train in these settings, while still achieving diverse solutions (discussed further below).

**Evaluation of Diversity.** Average ranking in terms of performance of the $\Delta$-spread metric for all algorithms (Table 5) shows that our approach gives more diverse solutions than all the other baselines including in the **RE** setting. These results highlight the importance of having a diversity constraint in the training procedure for the classifier through the data selection procedure (Appendix E.3).

Across experiments, we find that our approach has competitive performance in terms of hypervolume (convergence) while being better in terms of the $\Delta$-spread metric (diversity) than the baselines.

## 5.3 Ablation Studies

We also study the impact of choice of two important aspects: the guidance-weight hyperaprameter $w$, and the role of diversity criteria in training the preference classifier.

Results for various choices of $w$ for ZDT subtask are provided in Fig. 4a. With $w = 0$ (no preference guidance), the results are lower, as expected. Increasing the guidance weight generally results in better hypervolume at the slight cost of diversity. This indicates that if diversity is more important in the resulting designs, extreme values of $w$ are less preferable.

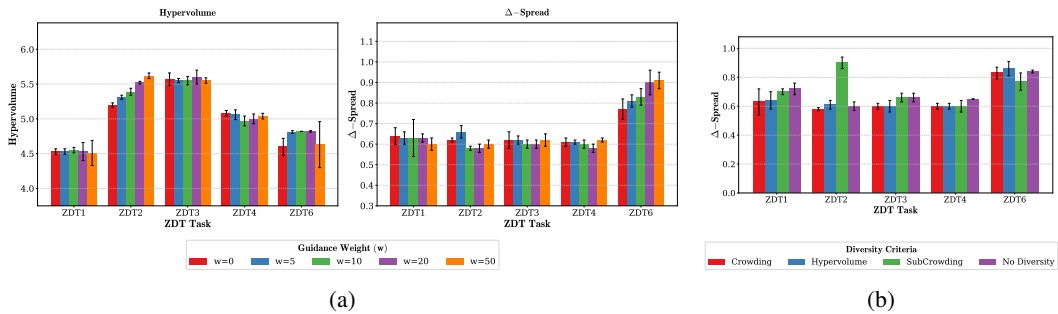

Figure 4: Ablation study of (a) different guidance weights $w$ on both hypervolume and $\Delta$-spread metrics (b) various diversity criteria (Crowding, SubCrowding, No Diversity and Hypervolume) on the $\Delta$-spread metric. All evaluation done with ZDT subtask on 256 sampled designs across 5 random seeds.

We also perform ablation studies on the role of diversity criteria that is used to pick preference pairs for training the preference classifier. For an effective comparison, we also compare with hypervolume improvement, in addition to Crowding, SubCrowding and having no diversity criteria. In the hypervolume improvement diversity metric (indicated as just *Hypervolume*), we measure diversity of a point based on the change in hypervolume it brings when it is added to the sampled set of designs. Results for $\Delta$-spread metric for the ZDT subtask is given in Fig. 4b. We find that incorporating diversity criteria for preference generally results in better diversity of the resulting solutions. In addition, Crowding and SubCrowding gives better diversity than using hypervolume improvement.

Detailed results for all tasks are provided in Appendix E.1.

Table 5: Average ranking of the $\Delta$-spread metric obtained by different algorithms on both synthetic and RE tasks. Detailed results are provided in Appendix E.

| Method | Synthetic | RE |
|---|---|---|
| MultiHead-GradNorm | 7.12 | 8.27 |
| MultiHead-PcGrad | 5.92 | 7.0 |
| MultiHead | 8.83 | 7.2 |
| MultipleModels-COM | 6.5 | 5.47 |
| MultipleModels-ICT | 6.5 | 5.87 |
| MultipleModels-IOM | 6.42 | 6.8 |
| MultipleModels-RoMA | 5.92 | 6.4 |
| MultipleModels-TriMentoring | 6.0 | 4.6 |
| MultipleModels | 9.25 | 7.53 |
| ParetoFlow | 10.0 | 9.0 |
| **PGD-MOO + Pruning (Ours)** | **2.67** | **4.0** |
| **PGD-MOO (Ours)** | 2.83 | **4.28** |

## 6 Conclusion

In this work, we presented a novel classifier-guided diffusion approach for offline multi-objective optimization (MOO). Our method leverages a preference model that predicts Pareto dominance between pairs of inputs, incorporating diversity considerations to ensure that designs on the same Pareto front are well-distributed. Empirical results show that our technique performs competitively in terms of convergence to the true Pareto front, while also generating a diverse set of solutions.

**Limitations.** A key limitation of our current approach is that it relies solely on dominance information rather than the individual function values of the objectives. Consequently, it does not allow fine-grained control over trade-offs among different objectives, which can be important if a practitioner needs to emphasize or de-emphasize specific objective values.

**Future Directions.** One promising extension would be to integrate additional guidance signals, such as the actual function values, enabling a more preference-based form of MOO. This would allow users to explicitly prioritize certain objectives over others or specify desired performance ranges. Another avenue for future work is combining forward (surrogate-based) and inverse (generative) approaches, where candidates proposed by the generative model are iteratively refined using surrogate models.

## Acknowledgements

This work was supported by a CIFAR Catalyst grant, the Helmholtz Foundation Model Initiative and the Helmholtz Association. The authors also acknowledge the Gauss Centre for Supercomputing e.V. (www.gauss-centre.eu) for funding this project by providing computing time on the GCS Supercomputer JUQUEEN [37] at Jülich Supercomputing Centre (JSC). BEE was funded in part by grants from the Parker Institute for Cancer Immunology (PICI), the Chan-Zuckerberg Institute (CZI), NIH NHGRI R01 HG012967, and NIH NHGRI R01 HG013736. BEE is a CIFAR Fellow in the Multiscale Human Program. Syrine Belakaria is supported by the Stanford Data Science fellowship.

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

# Appendix: Preference-Guided Diffusion for Multi-Objective Offline Optimization

## A  Experimental Setting

### A.1  Metrics

We provide computational details regarding the metrics used in the work - hypervolume and $\Delta$-spread.

- Hypervolume: Hypervolume is a common metric which is used in Multi-Objective Optimization to measure the quality of the solutions. It does not require access to the Pareto-Front. It instead relies on a reference point, which is any point in the input space that is worse than all points in the solution set across every objective. Given this reference point $\boldsymbol{x}_{\text{ref}}$, hypervolume is computed as the area enclosed by hyperrectangle from every point in the solution set $S$ to $\boldsymbol{x}_{\text{ref}}$:

$$\text{HV}(S) := \int \bigcup_{\boldsymbol{x} \in S} [\boldsymbol{x}_1, \boldsymbol{x}_{\text{ref}_1}] \times [\boldsymbol{x}_2, \boldsymbol{x}_{\text{ref}_2}] \times \cdots \times [\boldsymbol{x}_m, \boldsymbol{x}_{\text{ref}_m}] \, d\lambda \tag{8}$$

where $\lambda$ is the Lebesgue measure and $[\cdot, \cdot]$ corresponds to a hyperrectangle between the two points. While relatively easier for 2 or 3 objective problems, computing hypervolume is more involved for solutions with higher number of objectives.

- $\Delta$-Spread: $\Delta$-Spread [12, 51] measures the uniformity of solutions which belong to the same front. In this work, we take into account the extreme points of the predicted front to calculate the $\Delta$-spread of a solution set $S$ as follows:

$$\Delta(S) := \frac{\sum_{i=1}^{m} \min_y ||y^{i,*} - y|| + \sum_{k=1}^{|S|-1} \left[ d^c(y^k, S/\{y^k\}) - \hat{d}^c \right]}{\sum_{i=1}^{m} \min_y ||y^{i,*} - y|| + (|S| - 1)\hat{d}^c} \tag{9}$$

where $\min_y ||y^{i,*} - y||$ is the distance between the extreme points of the set $S$ and the corresponding extreme points in the Pareto front. If the Pareto front is not known, this quantity is evaluated to be zero. $d^c(y^k, S/\{y^k\})$ corresponds to the distance between consecutive points of set $S$, sorted according to objective values of one of the objectives, and $\hat{d}^c$ is the mean of $d^c$ across all elements of set $S$.

## B  Dataset Details

We provide further details of the datasets used in our experiments for both Synthetic and RE set of tasks in Tables 6 and 7. All the datasets are directly taken from [45] as well as the evaluation of hypervolume computation and the needed reference points.

Table 6: Dataset information and reference point for hypervolume computation **Synthetic** set of tasks.

| Name | $d$ | $m$ | Pareto Front Shape | Reference Point |
|------|-----|-----|--------------------|-----------------|
| DTLZ1 | 7 | 3 | Linear | (558.21, 552.30, 568.36) |
| DTLZ2 | 10 | 3 | Concave | (2.77, 2.78, 2.93) |
| DTLZ3 | 10 | 3 | Concave | (1703.72, 1605.54, 1670.48) |
| DTLZ4 | 10 | 3 | Concave | (3.03, 2.83, 2.78) |
| DTLZ5 | 10 | 3 | Concave (2d) | (2.65, 2.61, 2.70) |
| DTLZ6 | 10 | 3 | Concave (2d) | (9.80, 9.78, 9.78) |
| DTLZ7 | 10 | 3 | Disconnected | (1.10, 1.10, 33.43) |
| ZDT1 | 30 | 2 | Convex | (1.10, 8.58) |
| ZDT2 | 30 | 2 | Concave | (1.10, 9.59) |
| ZDT3 | 30 | 2 | Disconnected | (1.10, 8.74) |
| ZDT4 | 10 | 2 | Convex | (1.10, 300.42) |
| ZDT6 | 10 | 2 | Concave | (1.07, 10.27) |

Table 7: Dataset information and reference point for hypervolume computation for **RE** set of tasks.

| Name | $d$ | $m$ | Pareto Front Shape | Reference Point |
|---|---|---|---|---|
| RE21 (Four bar truss design) | 4 | 2 | Convex | (3144.44, 0.05) |
| RE22 (Reinforced concrete beam design) | 3 | 2 | Mixed | (829.08, 2407217.25) |
| RE23 (Pressure vessel design) | 4 | 2 | Mixed, Disconnected | (713710.88, 1288669.78) |
| RE24 (Hatch cover design) | 2 | 2 | Convex | (5997.83, 43.67) |
| RE25 (Coil compression spring design) | 3 | 2 | Mixed, Disconnected | (124.79, 10038735.00) |
| RE31 (Two bar truss design) | 3 | 3 | Unknown | (808.85, 6893375.82, 6793450.00) |
| RE32 (Welded beam design) | 4 | 3 | Unknown | (290.66, 16552.46, 388265024.00) |
| RE33 (Disc brake design) | 4 | 3 | Unknown | (8.01, 8.84, 2343.30) |
| RE34 (Vehicle crashworthiness design) | 5 | 3 | Unknown | (1702.52, 11.68, 0.26) |
| RE35 (Speed reducer design) | 7 | 3 | Unknown | (7050.79, 1696.67, 397.83) |
| RE36 (Gear train design) | 4 | 3 | Concave, Disconnected | (10.21, 60.00, 0.97) |
| RE37 (Rocket injector design) | 4 | 3 | Unknown | (0.99, 0.96, 0.99) |
| RE41 (Car side impact design) | 7 | 4 | Unknown | (42.65, 4.43, 13.08, 13.45) |
| RE42 (Conceptual marine design) | 6 | 4 | Unknown | (-26.39, 19904.90, 28546.79, 14.98) |
| RE61 (Water resource planning) | 3 | 6 | Unknown | (83060.03, 1350.00, 2853469.06, 16027067.60, 357719.74, 99660.36) |

Table 8: Dataset information and reference point for **MO-NAS** set of tasks.

| Name | Search space | $d$ | $m$ | Reference Point |
|---|---|---|---|---|
| C-10/MOP1 | NAS-Bench-101 | 26 | 2 | $(3.49 \times 10^{-1}, 3.14 \times 10^7)$ |
| C-10/MOP2 | NAS-Bench-101 | 26 | 3 | $(9.05 \times 10^{-1}, 3.05 \times 10^7, 8.97 \times 10^0)$ |
| C-10/MOP3 | NATS | 5 | 3 | $(2.31 \times 10^1, 7.14 \times 10^{-1}, 2.74 \times 10^2)$ |
| C-10/MOP4 | NATS | 5 | 4 | $(2.31 \times 10^1, 7.14 \times 10^{-1}, 2.74 \times 10^2, 2.12 \times 10^{-2})$ |
| C-10/MOP5 | NAS-Bench-201 | 6 | 5 | $(9.03 \times 10^1, 1.53 \times 10^0, 2.20 \times 10^2, 1.17 \times 10^1, 4.88 \times 10^1)$ |
| C-10/MOP6 | NAS-Bench-201 | 6 | 6 | $(9.03 \times 10^1, 1.53 \times 10^0, 2.20 \times 10^2, 1.05 \times 10^1, 2.23 \times 10^0, 2.76 \times 10^1)$ |
| C-10/MOP7 | NAS-Bench-201 | 6 | 8 | $(9.03 \times 10^1, 1.53 \times 10^0, 2.20 \times 10^2, 1.17 \times 10^1, 4.88 \times 10^1, 1.05 \times 10^1, 2.23 \times 10^0, 2.76 \times 10^1)$ |
| C-10/MOP8 | DARTS | 32 | 2 | $(2.61 \times 10^{-1}, 1.55 \times 10^6)$ |
| C-10/MOP9 | DARTS | 32 | 3 | $(4.85 \times 10^{-2}, 3.92 \times 10^5)$ |
| IN-1K/MOP1 | ResNet50 | 25 | 2 | $(2.81 \times 10^{-1}, 3.95 \times 10^7)$ |
| IN-1K/MOP2 | ResNet50 | 25 | 2 | $(2.80 \times 10^{-1}, 1.15 \times 10^{10})$ |
| IN-1K/MOP3 | ResNet50 | 25 | 3 | $(2.81 \times 10^{-1}, 3.87 \times 10^7, 1.26 \times 10^{10})$ |
| IN-1K/MOP4 | Transformer | 34 | 2 | $(1.83 \times 10^1, 7.25 \times 10^7)$ |
| IN-1K/MOP5 | Transformer | 34 | 3 | $(1.83 \times 10^1, 1.49 \times 10^{10})$ |
| IN-1K/MOP6 | Transformer | 34 | 3 | $(1.83 \times 10^1, 7.10 \times 10^7, 1.48 \times 10^{10})$ |
| IN-1K/MOP7 | MNV3 | 21 | 2 | $(2.64 \times 10^{-1}, 9.98 \times 10^6)$ |
| IN-1K/MOP8 | MNV3 | 21 | 3 | $(2.65 \times 10^{-1}, 1.00 \times 10^7, 1.34 \times 10^9)$ |
| IN-1K/MOP9 | MNV3 | 21 | 4 | $(2.65 \times 10^{-1}, 1.03 \times 10^7, 1.31 \times 10^9, 6.30 \times 10^1)$ |

## C  Computational Resources

All the experiments are run on an NVIDIA A100 GPU. Our proposed approach takes on average of 1300 seconds for a 2 objective task of 30 dimensions. Approaches like MultipleModels' walltime is directly proportional to the number of objectives. Consequently, Paretoflow [48], which relies on MultipleModels also scales poorly wrt number of objectives. Overall, our approach takes roughly 56 GPU hours to train and test on all the datasets for 5 seeds. A representative runtime of our approach as well as the baselines is given in Table 9.

## D  Broader Impact Statement

This paper addresses the problem of offline optimization with multiple objectives. Advances in the problem being addressed are impactful for discovering new drugs for curing diseases, discovering new materials with certain physical properties, to name a few. The authors do not forsee any negative societal impacts of this work beyond what might be enabled due to general advancements in machine learning.

Table 9: Runtime (in seconds, for training) of different approaches on two representative tasks. Both tasks have 60k samples in total. The runtime of PGD mainly depends on the dimensionality of the problem (for RE 61 it is 3 and ZDT1 is 30) while for approaches like MultiHead and ParetoFlow it is proportional to the number of objectives (for RE 61 it is 6 and for ZDT1 it is 2), since they require training a surrogate model for each objective. We expect these runtimes to be the same if the dimensionality (or correspondingly the number of objectives) for other tasks are the same. Sampling times for all these approaches are fairly fast (at most 30 sec for 256 samples).

| Method | ZDT1 | RE61 |
|---|---|---|
| **MultiHead** | 450 | 470 |
| **MultipleModels** | 920 | 2820 |
| **ParetoFlow** | 1200 | 3100 |
| **PGD-MOO (Ours)** | 1300 | 930 |

# E    Additional Results

## E.1    Ablation Study Full Results

In addition to the main results, we perform evaluation of the impact of guidance weight $w$ on the resulting designs. Tables 10, 12 and 14 presents results of hypervolume evaluated on 256 sampled designs. Tables 11, 13 and 15 presents the corresponding diversity evaluation with $\Delta$-spread metric.

In addition to the above evaluation, we also evaluate the impact of the incorporated diversity criteria for selecting the preference pairs for training the classifer. In addition to *Crowding* and *SubCrowding*, we also consider not incorporating any diversity (mentioned as *No diversity*), as well as computing the improvement in hypervolume due to specific design in the sampled designs (mentioend as just *Hypervolume*). Hypervolume improvement due to a specific sample can be computed as the difference between hypervolume due to the entire set and the hypervolume due to the entire set except for this specific sample.

Results for different evaluation criteria are provided in Tables 16, 18 and 20 for hypervolume, as well as in Tables 17, 19 and 21 for diversity.

Table 10: Hypervolume results with 256 sampled designs of different guidance weights $w$ for DTLZ subtasks (part of **synthetic** task).

| Guidance Weight ($w$) | DTLZ1 | DTLZ2 | DTLZ3 | DTLZ4 | DTLZ5 | DTLZ6 | DTLZ7 |
|---|---|---|---|---|---|---|---|
| $w = 0$ | 10.64±0.00 | 10.53±0.02 | 10.61±0.02 | 10.66±0.01 | 10.07±0.02 | 10.15±0.05 | 9.48±0.14 |
| $w = 5$ | 10.65±0.00 | 10.54±0.02 | 10.63±0.00 | 10.64±0.01 | 10.08±0.01 | 10.17±0.02 | 9.64±0.14 |
| $w = 10$ | 10.65±0.00 | 10.56±0.00 | 10.63±0.00 | 10.65±0.01 | 10.05±0.01 | 10.17±0.04 | 9.76±0.14 |
| $w = 20$ | 10.64±0.00 | 10.55±0.01 | 10.62±0.01 | 10.64±0.01 | 10.09±0.02 | 10.16±0.04 | 9.81±0.23 |
| $w = 50$ | 10.62±0.01 | 10.55±0.01 | 10.63±0.00 | 10.63±0.01 | 10.09±0.02 | 10.16±0.09 | 9.80±0.36 |

Table 11: Diversity evaluation results with 256 sampled designs of different guidance weights $w$ for DTLZ subtasks (part of **synthetic** task).

| Guidance Weight ($w$) | DTLZ1 | DTLZ2 | DTLZ3 | DTLZ4 | DTLZ5 | DTLZ6 | DTLZ7 |
|---|---|---|---|---|---|---|---|
| $w = 0$ | 0.54±0.02 | 0.54±0.02 | 0.48±0.01 | 1.66±0.03 | 0.50±0.02 | 0.64±0.01 | 0.52±0.02 |
| $w = 5$ | 0.59±0.04 | 0.54±0.03 | 0.48±0.03 | 1.62±0.04 | 0.52±0.04 | 0.62±0.02 | 0.65±0.04 |
| $w = 10$ | 0.61±0.04 | 0.56±0.02 | 0.47±0.02 | 1.62±0.03 | 0.54±0.02 | 0.62±0.02 | 0.70±0.05 |
| $w = 20$ | 0.61±0.05 | 0.54±0.03 | 0.47±0.01 | 1.53±0.02 | 0.50±0.03 | 0.60±0.03 | 0.73±0.07 |
| $w = 50$ | 0.60±0.01 | 0.56±0.02 | 0.49±0.03 | 1.49±0.05 | 0.53±0.03 | 0.60±0.02 | 0.66±0.06 |

Table 12: Hypervolume results with 256 sampled designs of different guidance weights $w$ for ZDT subtasks (part of **synthetic** task).

| Guidance Weight ($w$) | ZDT1 | ZDT2 | ZDT3 | ZDT4 | ZDT6 |
|---|---|---|---|---|---|
| $w = 0$ | 4.53±0.04 | 5.20±0.03 | 5.57±0.09 | 5.08±0.04 | 4.60±0.12 |
| $w = 5$ | 4.53±0.04 | 5.31±0.03 | 5.55±0.03 | 5.06±0.07 | 4.81±0.02 |
| $w = 10$ | 4.55±0.04 | 5.39±0.05 | 5.55±0.06 | 4.97±0.07 | 4.82±0.00 |
| $w = 20$ | 4.53±0.13 | 5.52±0.02 | 5.60±0.10 | 5.00±0.07 | 4.82±0.01 |
| $w = 50$ | 4.51±0.18 | 5.62±0.04 | 5.55±0.04 | 5.04±0.04 | 4.63±0.33 |

Table 13: Diversity evaluation results with 256 sampled designs of different guidance weights $w$ in ZDT subtasks (part of **synthetic** task).

| Guidance Weight ($w$) | ZDT1 | ZDT2 | ZDT3 | ZDT4 | ZDT6 |
|---|---|---|---|---|---|
| $w = 0$ | 0.64±0.04 | 0.62±0.01 | 0.62±0.04 | 0.61±0.02 | 0.77±0.05 |
| $w = 5$ | 0.63±0.03 | 0.66±0.03 | 0.62±0.02 | 0.61±0.01 | 0.81±0.03 |
| $w = 10$ | 0.63±0.09 | 0.58±0.01 | 0.60±0.02 | 0.60±0.02 | 0.83±0.04 |
| $w = 20$ | 0.63±0.02 | 0.58±0.02 | 0.60±0.02 | 0.58±0.02 | 0.90±0.06 |
| $w = 50$ | 0.60±0.03 | 0.60±0.02 | 0.62±0.03 | 0.62±0.01 | 0.91±0.04 |

Table 14: Hypervolume results with 256 sampled designs of different guidance weights $w$ for various **RE** tasks.

| Guidance Weight ($w$) | RE21 | RE24 | RE25 | RE35 | RE41 |
|---|---|---|---|---|---|
| $w = 0$ | 4.38±0.07 | 4.84±0.00 | 4.84±0.00 | 10.42±0.06 | 18.98±0.25 |
| $w = 5$ | 4.43±0.05 | 4.84±0.00 | 4.84±0.00 | 10.37±0.06 | 19.00±0.26 |
| $w = 10$ | 4.46±0.04 | 4.83±0.00 | 4.84±0.00 | 10.27±0.11 | 19.17±0.30 |
| $w = 20$ | 4.45±0.05 | 4.84±0.00 | 4.84±0.00 | 10.16±0.15 | 18.40±0.84 |
| $w = 50$ | 4.44±0.05 | 4.83±0.00 | 4.84±0.00 | 9.55±0.54 | 17.67±0.96 |

Table 15: Diversity evaluation results with 256 sampled designs of different guidance weights $w$ for various **RE** tasks.

| Guidance Weight ($w$) | RE21 | RE24 | RE25 | RE35 | RE41 |
|---|---|---|---|---|---|
| $w = 0$ | 0.60±0.03 | 1.15±0.07 | 1.05±0.05 | 0.70±0.03 | 0.42±0.02 |
| $w = 5$ | 0.61±0.04 | 1.20±0.09 | 1.08±0.05 | 0.81±0.12 | 0.45±0.02 |
| $w = 10$ | 0.61±0.02 | 1.20±0.05 | 1.19±0.08 | 1.00±0.12 | 0.47±0.04 |
| $w = 20$ | 0.61±0.03 | 1.18±0.03 | 1.21±0.06 | 0.83±0.11 | 0.49±0.04 |
| $w = 50$ | 0.63±0.02 | 1.17±0.08 | 1.21±0.06 | 0.82±0.09 | 0.62±0.19 |

Table 16: Hypervolume results with 256 sampled designs when specific diversity criterion is used for training the preference classifier. Results evaluated with $w = 10$ on DTLZ subtasks (part of **synthetic** tasks).

| Diversity Criteria | DTLZ1 | DTLZ2 | DTLZ3 | DTLZ4 | DTLZ5 | DTLZ6 | DTLZ7 |
|---|---|---|---|---|---|---|---|
| Crowding | 10.65±0.00 | 10.56±0.00 | 10.63±0.00 | 10.65±0.01 | 10.05±0.01 | 10.17±0.04 | 9.76±0.14 |
| Hypervolume | 10.64±0.00 | 10.55±0.01 | 10.63±0.00 | 10.64±0.00 | 10.06±0.02 | 10.13±0.01 | 9.72±0.09 |
| SubCrowding | 10.64±0.00 | 10.56±0.01 | 10.62±0.01 | 10.65±0.02 | 10.08±0.01 | 10.10±0.07 | 9.49±0.05 |
| No diversity | 10.64±0.00 | 10.56±0.00 | 10.63±0.00 | 10.64±0.01 | 10.06±0.00 | 10.13±0.00 | 9.66±0.00 |

## E.2 Visualization of the Sampled Designs

We provide visualization of the sampled designs from our model on 2-objective problems in Fig. 5.

Table 17: Diversity evaluation results with 256 sampled designs when specific diversity criterion is used for training the preference classifier. Results evaluated with $w = 10$ on DTLZ subtasks (part of **synthetic** tasks).

| Diversity Criteria | DTLZ1 | DTLZ2 | DTLZ3 | DTLZ4 | DTLZ5 | DTLZ6 | DTLZ7 |
|---|---|---|---|---|---|---|---|
| Crowding | 0.61±0.04 | 0.56±0.02 | 0.47±0.02 | 1.62±0.03 | 0.54±0.02 | 0.62±0.02 | 0.70±0.05 |
| Hypervolume | 0.58±0.00 | 0.54±0.03 | 0.49±0.01 | 1.63±0.04 | 0.53±0.02 | 0.63±0.02 | 0.76±0.07 |
| SubCrowding | 0.55±0.03 | 0.54±0.02 | 0.48±0.03 | 1.68±0.04 | 0.51±0.02 | 0.65±0.02 | 0.58±0.02 |
| No diversity | 0.58±0.00 | 0.55±0.00 | 0.48±0.00 | 1.56±0.03 | 0.53±0.00 | 0.63±0.00 | 0.68±0.00 |

Table 18: Hypervolume results with 256 sampled designs when specific diversity criterion is used for training the preference classifier. Results evaluated with $w = 10$ on ZDT subtasks (part of **synthetic** tasks).

| Diversity Criteria | ZDT1 | ZDT2 | ZDT3 | ZDT4 | ZDT6 |
|---|---|---|---|---|---|
| Crowding | 4.55±0.04 | 5.39±0.05 | 5.55±0.06 | 4.97±0.07 | 4.82±0.00 |
| Hypervolume | 4.49±0.12 | 5.40±0.05 | 5.59±0.09 | 5.01±0.11 | 4.82±0.01 |
| SubCrowding | 4.54±0.07 | 5.24±0.04 | 5.60±0.03 | 5.08±0.08 | 4.59±0.07 |
| No diversity | 4.59±0.08 | 5.46±0.03 | 5.58±0.00 | 5.21±0.00 | 4.74±0.00 |

Table 19: Diversity evaluation results with 256 sampled designs when specific diversity criterion is used for training the preference classifier. Results evaluated with $w = 10$ on ZDT subtasks (part of **synthetic** tasks).

| Diversity Criteria | ZDT1 | ZDT2 | ZDT3 | ZDT4 | ZDT6 |
|---|---|---|---|---|---|
| Crowding | 0.63±0.09 | 0.58±0.01 | 0.60±0.02 | 0.60±0.02 | 0.83±0.04 |
| Hypervolume | 0.64±0.06 | 0.61±0.03 | 0.60±0.04 | 0.60±0.02 | 0.86±0.05 |
| SubCrowding | 0.70±0.02 | 0.90±0.04 | 0.66±0.03 | 0.60±0.04 | 0.77±0.06 |
| No diversity | 0.72±0.04 | 0.60±0.03 | 0.66±0.03 | 0.65±0.00 | 0.84±0.01 |

Table 20: Hypervolume results with 256 sampled designs when specific diversity criterion is used for training the preference classifier. Results evaluated with $w = 10$ on **RE** tasks.

| Diversity Criteria | RE21 | RE24 | RE25 | RE35 | RE41 |
|---|---|---|---|---|---|
| Crowding | 4.46±0.04 | 4.83±0.00 | 4.84±0.00 | 10.27±0.11 | 19.17±0.30 |
| Hypervolume | 4.41±0.03 | 4.83±0.00 | 4.84±0.00 | 10.36±0.04 | 19.09±0.00 |
| SubCrowding | 4.39±0.05 | 4.83±0.00 | 4.84±0.00 | 10.41±0.05 | 19.02±0.18 |
| No diversity | 4.42±0.00 | 4.83±0.00 | 4.84±0.00 | 10.35±0.00 | 19.09±0.00 |

Table 21: Diversity evaluation results with 256 sampled designs when specific diversity criterion is used for training the preference classifier. Results evaluated with $w = 10$ on **RE** tasks.

| Diversity Criteria | RE21 | RE24 | RE25 | RE35 | RE41 |
|---|---|---|---|---|---|
| Crowding | 0.61±0.02 | 1.20±0.05 | 1.19±0.08 | 1.00±0.12 | 0.47±0.04 |
| Hypervolume | 0.62±0.02 | 1.16±0.09 | 1.14±0.09 | 0.99±0.08 | 0.47±0.03 |
| SubCrowding | 0.60±0.01 | 1.17±0.05 | 1.12±0.06 | 0.70±0.03 | 0.43±0.02 |
| No diversity | 0.62±0.01 | 1.16±0.06 | 1.39±0.06 | 0.90±0.02 | 0.50±0.04 |

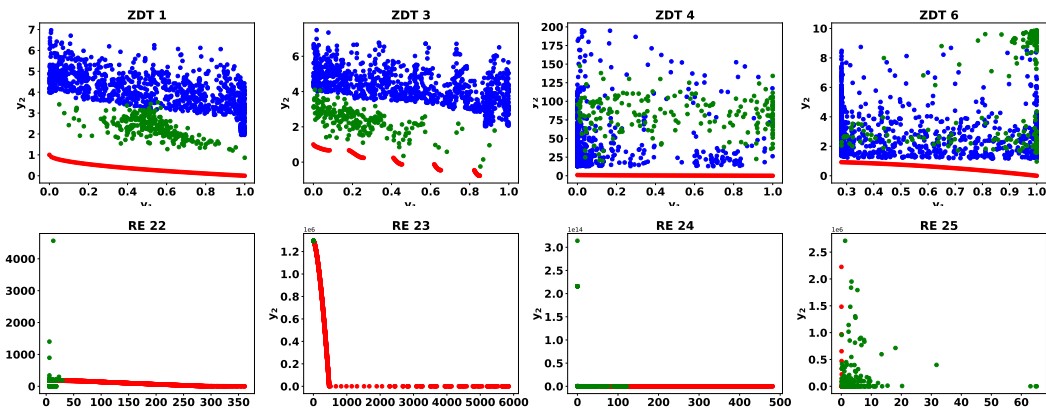

Figure 5: Plot of the samples from diffusion model (in green) on different 2-objective tasks. Top row shows results for synthetic set of benchmarks, while the bottom row shows results for RE engineering suite. Blue dots correspond to training data and red dot corresponds to the true Pareto front. The blue dots are omitted for the bottom row for clarity. Results for ZDT2 is available in Fig. 3.

Table 22: Evaluation of hypervolume with 256 sampled designs on subsets of the **RE** task. Results are averaged over 5 different random seeds.

| Method | RE21 | RE22 | RE23 | RE24 | RE25 | RE31 | RE32 | RE33 |
|---|---|---|---|---|---|---|---|---|
| D(best) | 4.1 | 4.78 | 4.75 | 4.6 | 4.79 | 10.6 | 10.56 | 10.56 |
| MultiHead-GradNorm | $4.28 \pm 0.39$ | $4.7 \pm 0.44$ | $3.77 \pm 1.12$ | $3.65 \pm 0.82$ | $4.52 \pm 0.5$ | $10.6 \pm 0.1$ | $10.54 \pm 0.15$ | $10.03 \pm 1.5$ |
| MultiHead-PcGrad | $4.59 \pm 0.01$ | $4.73 \pm 0.36$ | $4.84 \pm 0.0$ | $4.15 \pm 0.66$ | $4.78 \pm 0.14$ | $10.64 \pm 0.01$ | $10.63 \pm 0.01$ | $10.59 \pm 0.03$ |
| MultiHead | $4.6 \pm 0.0$ | $4.84 \pm 0.0$ | $4.84 \pm 0.01$ | $4.73 \pm 0.2$ | $4.74 \pm 0.2$ | $10.65 \pm 0.0$ | $10.6 \pm 0.05$ | $10.62 \pm 0.0$ |
| MultipleModels-COM | $4.38 \pm 0.09$ | $4.84 \pm 0.0$ | $4.84 \pm 0.0$ | $4.73 \pm 0.2$ | $4.83 \pm 0.01$ | $10.64 \pm 0.01$ | $10.64 \pm 0.01$ | $10.61 \pm 0.0$ |
| MultipleModels-ICT | $4.6 \pm 0.0$ | $4.84 \pm 0.0$ | $4.45 \pm 0.02$ | $4.83 \pm 0.01$ | $4.84 \pm 0.0$ | $10.65 \pm 0.0$ | $10.64 \pm 0.0$ | $10.62 \pm 0.0$ |
| MultipleModels-IOM | $4.58 \pm 0.02$ | $4.84 \pm 0.0$ | $4.83 \pm 0.01$ | $4.72 \pm 0.11$ | $4.83 \pm 0.01$ | $10.65 \pm 0.0$ | $10.65 \pm 0.0$ | $10.62 \pm 0.0$ |
| MultipleModels-RoMA | $4.57 \pm 0.0$ | $4.61 \pm 0.51$ | $4.83 \pm 0.01$ | $3.96 \pm 1.2$ | $4.83 \pm 0.01$ | $10.64 \pm 0.01$ | $10.64 \pm 0.0$ | $10.58 \pm 0.03$ |
| MultipleModels-TriMentoring | $4.6 \pm 0.0$ | $4.84 \pm 0.0$ | $4.84 \pm 0.0$ | $4.84 \pm 0.0$ | $4.84 \pm 0.0$ | $10.65 \pm 0.0$ | $10.62 \pm 0.01$ | $10.6 \pm 0.01$ |
| MultipleModels | $4.6 \pm 0.0$ | $4.84 \pm 0.0$ | $4.84 \pm 0.0$ | $4.83 \pm 0.01$ | $4.63 \pm 0.25$ | $10.65 \pm 0.0$ | $10.62 \pm 0.02$ | $10.62 \pm 0.0$ |
| ParetoFlow | $4.2 \pm 0.17$ | $4.86 \pm 0.01$ | - | - | - | $10.66 \pm 0.12$ | $10.61 \pm 0.0$ | $10.75 \pm 0.2$ |
| **PGD-MOO + Data Pruning (Ours)** | $4.42 \pm 0.04$ | $4.83 \pm 0.01$ | $4.84 \pm 0.0$ | $4.84 \pm 0.0$ | $4.84 \pm 0.0$ | $10.57 \pm 0.05$ | $10.64 \pm 0.0$ | $10.09 \pm 0.6$ |
| **PGD-MOO (Ours)** | $4.46 \pm 0.03$ | $4.84 \pm 0.0$ | $4.84 \pm 0.0$ | $4.84 \pm 0.0$ | $4.84 \pm 0.0$ | $10.6 \pm 0.01$ | $10.65 \pm 0.0$ | $10.51 \pm 0.04$ |

## E.3 Detailed Results

Table 27: Evaluation of the $\Delta$-spread metric with 256 sampled designs on ZDT subtask, part of the **synthetic** set of tasks. Results are averaged over 5 different random seeds. Lower values are better.

| Method | ZDT1 | ZDT2 | ZDT3 | ZDT4 | ZDT6 |
|---|---|---|---|---|---|
| MultiHead-GradNorm | $0.96 \pm 0.19$ | $0.94 \pm 0.16$ | $0.93 \pm 0.17$ | $0.79 \pm 0.15$ | $0.77 \pm 0.16$ |
| MultiHead-PcGrad | $0.83 \pm 0.1$ | $0.98 \pm 0.13$ | $0.79 \pm 0.03$ | $0.67 \pm 0.04$ | $0.82 \pm 0.11$ |
| MultiHead | $1.13 \pm 0.08$ | $1.04 \pm 0.05$ | $0.83 \pm 0.06$ | $0.68 \pm 0.03$ | $1.22 \pm 0.07$ |
| MultipleModels-COM | $0.89 \pm 0.04$ | $0.79 \pm 0.11$ | $0.83 \pm 0.05$ | $0.64 \pm 0.03$ | $1.0 \pm 0.09$ |
| MultipleModels-ICT | $1.1 \pm 0.02$ | $1.01 \pm 0.07$ | $0.86 \pm 0.06$ | $0.69 \pm 0.07$ | $0.98 \pm 0.06$ |
| MultipleModels-IOM | $0.94 \pm 0.1$ | $0.9 \pm 0.05$ | $0.81 \pm 0.07$ | $0.73 \pm 0.04$ | $\mathbf{0.46 \pm 0.1}$ |
| MultipleModels-RoMA | $0.64 \pm 0.06$ | $0.92 \pm 0.1$ | $0.79 \pm 0.07$ | $0.69 \pm 0.02$ | $0.78 \pm 0.06$ |
| MultipleModels-TriMentoring | $0.86 \pm 0.03$ | $0.86 \pm 0.06$ | $0.9 \pm 0.04$ | $0.73 \pm 0.02$ | $0.78 \pm 0.06$ |
| MultipleModels | $1.07 \pm 0.06$ | $1.01 \pm 0.03$ | $0.84 \pm 0.03$ | $0.7 \pm 0.05$ | $1.19 \pm 0.04$ |
| ParetoFlow | $1.46 \pm 0.03$ | $1.19 \pm 0.1$ | $1.46 \pm 0.14$ | $1.31 \pm 0.1$ | $0.71 \pm 0.05$ |
| **PGD-MOO + DataPruning (Ours)** | $\mathbf{0.66 \pm 0.08}$ | $\mathbf{0.61 \pm 0.03}$ | $\mathbf{0.6 \pm 0.03}$ | $\mathbf{0.6 \pm 0.04}$ | $0.8 \pm 0.05$ |
| **PGD-MOO (Ours)** | $\mathbf{0.68 \pm 0.07}$ | $0.78 \pm 0.09$ | $0.65 \pm 0.03$ | $\mathbf{0.6 \pm 0.03}$ | $0.76 \pm 0.03$ |

The detailed results are given in Tables 22, 23 and 26 to 29.

Table 23: Evaluation of hypervolume with 256 sampled designs on subsets of the **RE** task. Results are averaged over 5 different random seeds.

| Method | RE34 | RE35 | RE36 | RE37 | RE41 | RE42 | RE61 |
|---|---|---|---|---|---|---|---|
| D(best) | 9.3 | 10.08 | 7.61 | 5.57 | 18.27 | 14.52 | 97.49 |
| MultiHead-GradNorm | $8.47 \pm 1.87$ | $9.76 \pm 1.3$ | $9.67 \pm 0.43$ | $5.67 \pm 1.41$ | $17.06 \pm 3.82$ | $18.77 \pm 2.99$ | $108.01 \pm 1.0$ |
| MultiHead-PcGrad | $10.11 \pm 0.0$ | $10.51 \pm 0.05$ | $10.17 \pm 0.08$ | $6.68 \pm 0.06$ | $20.66 \pm 0.1$ | $22.57 \pm 0.26$ | $108.54 \pm 0.23$ |
| MultiHead | $10.1 \pm 0.01$ | $10.49 \pm 0.07$ | $10.23 \pm 0.03$ | $6.67 \pm 0.05$ | $20.62 \pm 0.11$ | $22.38 \pm 0.35$ | $108.92 \pm 0.22$ |
| MultipleModels-COM | $9.96 \pm 0.09$ | $10.55 \pm 0.02$ | $9.82 \pm 0.35$ | $6.35 \pm 0.1$ | $20.37 \pm 0.06$ | $17.44 \pm 0.71$ | $107.99 \pm 0.48$ |
| MultipleModels-ICT | $10.1 \pm 0.0$ | $10.5 \pm 0.01$ | $10.29 \pm 0.03$ | $6.73 \pm 0.0$ | $20.58 \pm 0.04$ | $22.27 \pm 0.15$ | $108.68 \pm 0.27$ |
| MultipleModels-IOM | $10.11 \pm 0.01$ | $10.57 \pm 0.01$ | $10.29 \pm 0.04$ | $6.71 \pm 0.02$ | $20.66 \pm 0.05$ | $22.43 \pm 0.1$ | $107.71 \pm 0.5$ |
| MultipleModels-RoMA | $9.91 \pm 0.01$ | $10.53 \pm 0.03$ | $9.72 \pm 0.28$ | $6.67 \pm 0.02$ | $20.39 \pm 0.09$ | $21.41 \pm 0.37$ | $108.47 \pm 0.29$ |
| MultipleModels-TriMentoring | $10.08 \pm 0.02$ | $10.59 \pm 0.0$ | $9.64 \pm 1.42$ | $6.73 \pm 0.01$ | $20.68 \pm 0.04$ | $21.6 \pm 0.19$ | $108.61 \pm 0.29$ |
| MultipleModels | $10.11 \pm 0.0$ | $10.55 \pm 0.01$ | $10.24 \pm 0.03$ | $6.73 \pm 0.03$ | $20.77 \pm 0.08$ | $22.59 \pm 0.11$ | $108.96 \pm 0.06$ |
| ParetoFlow | $11.2 \pm 0.35$ | $11.12 \pm 0.02$ | $8.42 \pm 0.35$ | $6.55 \pm 0.59$ | $19.41 \pm 0.92$ | $20.35 \pm 5.31$ | $107.1 \pm 6.96$ |
| **PGD-MOO + Data Pruning (Ours)** | $9.15 \pm 0.11$ | $10.43 \pm 0.04$ | $9.48 \pm 0.33$ | $5.99 \pm 0.18$ | $19.37 \pm 0.15$ | $17.4 \pm 0.63$ | $103.04 \pm 1.71$ |
| **PGD-MOO (Ours)** | $9.39 \pm 0.16$ | $10.32 \pm 0.1$ | $9.37 \pm 0.17$ | $6.13 \pm 0.12$ | $19.31 \pm 0.46$ | $19.01 \pm 0.68$ | $105.02 \pm 1.14$ |

Table 24: Evaluation of hypervolume with 256 sampled designs on subsets of the **C10MOP** tasks. Results are averaged over 5 different random seeds.

| Approach | C10MOP1 | C10MOP2 | C10MOP3 | C10MOP4 | C10MOP5 | C10MOP6 | C10MOP7 | C10MOP8 | C10MOP9 |
|---|---|---|---|---|---|---|---|---|---|
| MultipleModels | $1.3858 \pm 0.0442$ | $1.3296 \pm 0.0300$ | $10.0132 \pm 0.2535$ | $21.8476 \pm 0.2264$ | $36.8597 \pm 2.7523$ | $95.0340 \pm 7.5757$ | $362.8160 \pm 18.0855$ | $5.2041 \pm 0.1013$ | $11.6359 \pm 0.4404$ |
| MultipleModels-COM | $*1.4702 \pm 0.0000*$ | $1.3321 \pm 0.0054$ | $11.0189 \pm 0.0365$ | $23.0747 \pm 0.0868$ | $49.7248 \pm 0.1496$ | $*107.0121 \pm 0.7802*$ | $480.4704 \pm 14.0997$ | $5.2402 \pm 0.0257$ | $13.9464 \pm 0.3804$ |
| MultipleModels-ICT | $1.3526 \pm 0.0150$ | $1.3234 \pm 0.0446$ | $9.9906 \pm 0.3720$ | $20.2256 \pm 0.7048$ | $38.9028 \pm 0.4978$ | $95.8324 \pm 6.6621$ | $334.2373 \pm 41.4124$ | $4.9339 \pm 0.0759$ | $12.4803 \pm 0.2973$ |
| MultipleModels-IOM | $1.3746 \pm 0.1341$ | $1.2992 \pm 0.0603$ | $10.5681 \pm 0.1584$ | $21.9670 \pm 0.3673$ | $47.6971 \pm 0.9712$ | $99.9001 \pm 3.1050$ | $436.2843 \pm 27.0213$ | $5.1168 \pm 0.0708$ | $13.7110 \pm 0.4607$ |
| MultipleModels-RoMA | $1.3660 \pm 0.0669$ | $1.3355 \pm 0.0251$ | $9.8492 \pm 0.3837$ | $21.4023 \pm 0.4962$ | $38.4872 \pm 3.4435$ | $93.0900 \pm 3.6038$ | $357.7016 \pm 41.3808$ | $5.0832 \pm 0.0801$ | $13.5530 \pm 0.2167$ |
| MultipleModels-TriMentoring | $1.4042 \pm 0.0336$ | $1.2416 \pm 0.0652$ | $10.0904 \pm 0.2997$ | $21.3896 \pm 1.0433$ | $40.7722 \pm 3.5499$ | $96.7069 \pm 4.5694$ | $408.3003 \pm 28.6059$ | $4.9647 \pm 0.0968$ | $12.7916 \pm 0.2853$ |
| MultiHead | $1.4559 \pm 0.0227$ | $1.2746 \pm 0.0588$ | $10.2391 \pm 0.2114$ | $21.2637 \pm 1.2549$ | $36.0405 \pm 3.3757$ | $98.8094 \pm 0.6955$ | $340.8008 \pm 60.0297$ | $5.0691 \pm 0.1096$ | $11.4813 \pm 0.5663$ |
| MultiHead-GradNorm | $1.4309 \pm 0.0155$ | $1.2636 \pm 0.0572$ | $10.0193 \pm 0.2474$ | $20.7159 \pm 1.0982$ | $30.5653 \pm 5.2836$ | $77.9574 \pm 11.9034$ | $306.7220 \pm 39.5226$ | $5.3211 \pm 0.1007$ | $12.9097 \pm 0.5352$ |
| MultiHead-PcGrad | $1.4601 \pm 0.0102$ | $1.3923 \pm 0.0787$ | $10.2801 \pm 0.2455$ | $21.4537 \pm 0.6093$ | $38.9828 \pm 0.5494$ | $88.2995 \pm 9.7358$ | $360.9641 \pm 30.9560$ | $4.9822 \pm 0.0631$ | $12.6775 \pm 0.3565$ |
| ParetoFlow | $1.4456 \pm 0.0171$ | $1.3437 \pm 0.0164$ | $10.6091 \pm 0.2478$ | $21.6532 \pm 0.6013$ | $48.6245 \pm 0.8522$ | $104.8361 \pm 1.9998$ | $463.2916 \pm 21.6113$ | $5.2958 \pm 0.0921$ | $*14.3905 \pm 0.2319*$ |
| **PGD-MOO + Data Pruning (Ours)** | $1.4817 \pm 0.0021$ | $1.3817 \pm 0.0138$ | $*11.1524 \pm 0.0165*$ | $*23.8769 \pm 0.1086*$ | $*49.8578 \pm 0.0266*$ | $106.6697 \pm 0.9115$ | $500.4782 \pm 1.7625$ | $5.5659 \pm 0.0156$ | $14.4528 \pm 0.1726$ |
| **PGD-MOO (Ours)** | $1.4511 \pm 0.0141$ | $*1.3598 \pm 0.0105*$ | $11.1826 \pm 0.0148$ | $23.9233 \pm 0.0821$ | $49.8683 \pm 0.0658$ | $107.2650 \pm 0.6335$ | $*499.1712 \pm 1.3745*$ | $*5.4807 \pm 0.0363*$ | $14.0768 \pm 0.2056$ |

Table 25: Evaluation of hypervolume with 256 sampled designs on the **IN1K** tasks. Results are averaged over 5 different random seeds.

| Approach | IN1KMOP1 | IN1KMOP2 | IN1KMOP3 | IN1KMOP4 | IN1KMOP5 | IN1KMOP6 | IN1KMOP7 | IN1KMOP8 | IN1KMOP9 |
|---|---|---|---|---|---|---|---|---|---|
| MultipleModels | $5.6231 \pm 0.0610$ | $6.0828 \pm 0.2808$ | $14.3020 \pm 0.0576$ | $4.1980 \pm 0.0303$ | $4.4503 \pm 0.0281$ | $10.6306 \pm 0.3082$ | $4.6675 \pm 0.1023$ | $9.1011 \pm 0.1883$ | $11.2394 \pm 0.3227$ |
| MultipleModels-COM | $5.4126 \pm 0.0963$ | $6.1518 \pm 0.0568$ | $14.6918 \pm 0.1703$ | $4.3015 \pm 0.0448$ | $4.5755 \pm 0.0157$ | $11.3625 \pm 0.2449$ | $5.1201 \pm 0.1738$ | $11.3815 \pm 0.0824$ | $14.9066 \pm 0.1649$ |
| MultipleModels-ICT | $5.6369 \pm 0.0287$ | $6.0640 \pm 0.2808$ | $14.3255 \pm 0.1601$ | $4.2440 \pm 0.0171$ | $4.3804 \pm 0.0231$ | $10.9187 \pm 0.0799$ | $4.9460 \pm 0.0873$ | $9.6489 \pm 0.2599$ | $11.6043 \pm 0.6310$ |
| MultipleModels-IOM | $5.5938 \pm 0.0890$ | $6.0892 \pm 0.2250$ | $14.7332 \pm 0.0914$ | $4.1076 \pm 0.4522$ | $4.6096 \pm 0.0408$ | $10.6794 \pm 1.0307$ | $5.1611 \pm 0.1263$ | $*11.1559 \pm 0.2388*$ | $*14.6504 \pm 0.6083*$ |
| MultipleModels-RoMA | $5.5776 \pm 0.0557$ | $6.1352 \pm 0.1875$ | $14.3641 \pm 0.2001$ | $4.1302 \pm 0.0364$ | $3.6911 \pm 0.0412$ | $8.0256 \pm 0.3621$ | $4.7364 \pm 0.1182$ | $9.0267 \pm 0.1139$ | $6.0159 \pm 0.3216$ |
| MultipleModels-TriMentoring | $5.4462 \pm 0.0192$ | $5.3899 \pm 0.0119$ | $14.2874 \pm 0.1749$ | $4.3775 \pm 0.0281$ | $4.0060 \pm 0.0317$ | $10.3430 \pm 0.0609$ | $5.0492 \pm 0.0850$ | $10.1604 \pm 0.3933$ | $11.0185 \pm 0.4408$ |
| MultiHead | $5.6206 \pm 0.0161$ | $5.8983 \pm 0.1535$ | $14.2471 \pm 0.1133$ | $4.1108 \pm 0.1259$ | $4.2844 \pm 0.0877$ | $10.3601 \pm 0.0467$ | $4.8340 \pm 0.0840$ | $9.2124 \pm 0.1680$ | $10.2314 \pm 0.9201$ |
| MultiHead-GradNorm | $5.1362 \pm 0.5426$ | $6.2062 \pm 0.0950$ | $14.0209 \pm 0.1754$ | $3.1755 \pm 0.0474$ | $4.2984 \pm 0.6297$ | $8.1613 \pm 0.7969$ | $4.6107 \pm 0.2090$ | $10.1800 \pm 0.6765$ | $11.4468 \pm 1.1444$ |
| MultiHead-PcGrad | $5.5797 \pm 0.0810$ | $6.0273 \pm 0.3497$ | $14.2122 \pm 0.0955$ | $4.2014 \pm 0.0729$ | $4.2996 \pm 0.0449$ | $10.8542 \pm 0.1305$ | $4.8260 \pm 0.3493$ | $8.7804 \pm 0.2875$ | $10.2910 \pm 0.0455$ |
| ParetoFlow | $5.3459 \pm 0.1106$ | $5.9646 \pm 0.1658$ | $14.3332 \pm 0.2594$ | $4.3161 \pm 0.0514$ | $4.6240 \pm 0.0506$ | $11.6062 \pm 0.0935$ | $4.8707 \pm 0.0423$ | $11.1389 \pm 0.1251$ | $14.4814 \pm 0.1094$ |
| **PGD-MOO + Data Pruning (Ours)** | $*5.7570 \pm 0.0664*$ | $6.4504 \pm 0.0201$ | $*14.7568 \pm 0.1310*$ | $*4.3928 \pm 0.0296*$ | $*4.6531 \pm 0.0426*$ | $*11.7936 \pm 0.1010*$ | $5.3022 \pm 0.0705$ | $10.6270 \pm 0.1074$ | $13.9399 \pm 0.2251$ |
| **PGD-MOO (Ours)** | $5.7638 \pm 0.0489$ | $*6.4305 \pm 0.0232*$ | $14.8635 \pm 0.1562$ | $4.4924 \pm 0.0105$ | $4.7465 \pm 0.0146$ | $11.9365 \pm 0.0626$ | $*5.1797 \pm 0.0780*$ | $10.7654 \pm 0.1653$ | $13.6189 \pm 0.3393$ |

Table 26: Evaluation of the $\Delta$-spread metric with 256 sampled designs on DTLZ subtask, part of the **synthetic** set of tasks. Results are averaged over 5 different random seeds. Lower values are better.

| Method | DTLZ1 | DTLZ2 | DTLZ3 | DTLZ4 | DTLZ5 | DTLZ6 | DTLZ7 |
|---|---|---|---|---|---|---|---|
| MultiHead-GradNorm | $0.61 \pm 0.03$ | $0.89 \pm 0.24$ | $0.88 \pm 0.29$ | $0.96 \pm 0.14$ | $0.75 \pm 0.1$ | $0.95 \pm 0.28$ | $1.2 \pm 0.18$ |
| MultiHead-PcGrad | $0.65 \pm 0.04$ | $0.73 \pm 0.05$ | $0.76 \pm 0.11$ | $1.08 \pm 0.2$ | $0.77 \pm 0.06$ | $0.87 \pm 0.06$ | $1.15 \pm 0.2$ |
| MultiHead | $0.86 \pm 0.08$ | $0.93 \pm 0.15$ | $0.93 \pm 0.16$ | $0.98 \pm 0.12$ | $0.92 \pm 0.14$ | $0.88 \pm 0.2$ | $0.93 \pm 0.11$ |
| MultipleModels-COM | $0.69 \pm 0.02$ | $0.85 \pm 0.15$ | $0.73 \pm 0.06$ | $1.09 \pm 0.12$ | $0.89 \pm 0.08$ | $1.15 \pm 0.15$ | $0.78 \pm 0.03$ |
| MultipleModels-ICT | $0.7 \pm 0.02$ | $0.79 \pm 0.02$ | $0.63 \pm 0.1$ | $0.92 \pm 0.03$ | $0.8 \pm 0.05$ | $0.91 \pm 0.06$ | $0.77 \pm 0.05$ |
| MultipleModels-IOM | $0.66 \pm 0.02$ | $0.96 \pm 0.18$ | $0.67 \pm 0.04$ | $1.23 \pm 0.24$ | $0.91 \pm 0.07$ | $1.11 \pm 0.09$ | $0.75 \pm 0.03$ |
| MultipleModels-RoMA | $0.62 \pm 0.03$ | $1.06 \pm 0.08$ | $0.89 \pm 0.11$ | $1.28 \pm 0.08$ | $0.93 \pm 0.13$ | $0.76 \pm 0.1$ | $0.69 \pm 0.03$ |
| MultipleModels-TriMentoring | $0.72 \pm 0.03$ | $0.91 \pm 0.1$ | $0.66 \pm 0.09$ | $0.95 \pm 0.07$ | $0.7 \pm 0.06$ | $0.8 \pm 0.09$ | $0.81 \pm 0.05$ |
| MultipleModels | $0.88 \pm 0.07$ | $1.11 \pm 0.26$ | $1.0 \pm 0.22$ | $0.93 \pm 0.13$ | $0.92 \pm 0.17$ | $1.0 \pm 0.24$ | $0.85 \pm 0.12$ |
| ParetoFlow | $0.82 \pm 0.02$ | $1.07 \pm 0.07$ | $0.68 \pm 0.09$ | $1.63 \pm 0.15$ | $1.04 \pm 0.06$ | $1.16 \pm 0.09$ | $0.84 \pm 0.09$ |
| **PGD-MOO + Data Pruning (Ours)** | $0.58 \pm 0.04$ | $0.52 \pm 0.03$ | $0.46 \pm 0.02$ | $1.66 \pm 0.04$ | $0.51 \pm 0.02$ | $0.63 \pm 0.02$ | $0.53 \pm 0.04$ |
| **PGD-MOO (Ours)** | $0.62 \pm 0.02$ | $0.5 \pm 0.03$ | $0.46 \pm 0.02$ | $1.6 \pm 0.04$ | $0.51 \pm 0.02$ | $0.61 \pm 0.03$ | $0.63 \pm 0.02$ |

Table 28: Evaluation of the $\Delta$-spread metric with 256 sampled designs on subsets of the **RE** task. Results are averaged over 5 different random seeds. Lower values are better.

| Method | RE21 | RE22 | RE23 | RE24 | RE25 | RE31 | RE32 | RE33 |
|---|---|---|---|---|---|---|---|---|
| MultiHead-GradNorm | $0.77 \pm 0.15$ | $1.61 \pm 0.37$ | $1.7 \pm 0.35$ | $0.98 \pm 0.4$ | $1.54 \pm 0.5$ | $0.91 \pm 0.17$ | $1.26 \pm 0.28$ | $0.92 \pm 0.14$ |
| MultiHead-PcGrad | $0.47 \pm 0.04$ | $1.8 \pm 0.22$ | $1.14 \pm 0.21$ | $1.25 \pm 0.4$ | $1.6 \pm 0.28$ | $1.05 \pm 0.25$ | $1.09 \pm 0.13$ | $0.91 \pm 0.14$ |
| MultiHead | $0.42 \pm 0.04$ | $1.43 \pm 0.22$ | $1.03 \pm 0.23$ | $1.13 \pm 0.19$ | $1.86 \pm 0.04$ | $0.94 \pm 0.14$ | $0.84 \pm 0.14$ | $1.16 \pm 0.05$ |
| MultipleModels-COM | $0.56 \pm 0.12$ | $1.22 \pm 0.42$ | $1.2 \pm 0.49$ | $1.08 \pm 0.23$ | $1.63 \pm 0.1$ | $1.24 \pm 0.17$ | $1.23 \pm 0.19$ | $0.87 \pm 0.07$ |
| MultipleModels-ICT | $0.38 \pm 0.02$ | $1.78 \pm 0.12$ | $0.84 \pm 0.07$ | $0.67 \pm 0.12$ | $1.54 \pm 0.04$ | $1.28 \pm 0.19$ | $0.91 \pm 0.09$ | $1.0 \pm 0.24$ |
| MultipleModels-IOM | $0.98 \pm 0.4$ | $1.84 \pm 0.08$ | $1.26 \pm 0.23$ | $1.58 \pm 0.24$ | $1.54 \pm 0.25$ | $1.07 \pm 0.27$ | $1.09 \pm 0.19$ | $0.99 \pm 0.06$ |
| MultipleModels-RoMA | $1.19 \pm 0.09$ | $1.57 \pm 0.56$ | $1.33 \pm 0.49$ | $1.22 \pm 0.24$ | $1.44 \pm 0.43$ | $1.39 \pm 0.27$ | $1.15 \pm 0.18$ | $0.89 \pm 0.06$ |
| MultipleModels-TriMentoring | $0.37 \pm 0.01$ | $1.73 \pm 0.12$ | $0.91 \pm 0.15$ | $0.56 \pm 0.03$ | $1.35 \pm 0.04$ | $1.17 \pm 0.14$ | $0.99 \pm 0.09$ | $0.85 \pm 0.03$ |
| MultipleModels | $0.37 \pm 0.02$ | $1.85 \pm 0.12$ | $0.82 \pm 0.46$ | $0.99 \pm 0.22$ | $1.72 \pm 0.36$ | $1.65 \pm 0.22$ | $0.9 \pm 0.26$ | $1.16 \pm 0.2$ |
| ParetoFlow | $1.5 \pm 0.12$ | $1.37 \pm 0.11$ | - | - | - | $1.66 \pm 0.03$ | $1.34 \pm 0.0$ | $1.07 \pm 0.11$ |
| **PGD-MOO + Data Pruning (Ours)** | $0.61 \pm 0.03$ | $1.29 \pm 0.08$ | $1.42 \pm 0.11$ | $1.13 \pm 0.01$ | $1.12 \pm 0.08$ | $1.34 \pm 0.28$ | $1.62 \pm 0.1$ | $0.83 \pm 0.17$ |
| **PGD-MOO (Ours)** | $0.61 \pm 0.03$ | $1.28 \pm 0.07$ | $1.08 \pm 0.06$ | $1.14 \pm 0.02$ | $1.17 \pm 0.07$ | $1.32 \pm 0.15$ | $1.59 \pm 0.04$ | $0.89 \pm 0.06$ |

Table 29: Evaluation of the $\Delta$-spread metric with 256 sampled designs on subsets of the **RE** task. Results are averaged over 5 different random seeds. Lower values are better.

| Method | RE34 | RE35 | RE36 | RE37 | RE41 | RE42 | RE61 |
|---|---|---|---|---|---|---|---|
| MultiHead-GradNorm | $0.96 \pm 0.23$ | $1.17 \pm 0.14$ | $1.19 \pm 0.19$ | $1.19 \pm 0.51$ | $1.13 \pm 0.5$ | $1.12 \pm 0.51$ | $0.72 \pm 0.05$ |
| MultiHead-PcGrad | $1.11 \pm 0.08$ | $0.99 \pm 0.13$ | $0.91 \pm 0.17$ | $0.76 \pm 0.04$ | $0.62 \pm 0.01$ | $0.9 \pm 0.08$ | $0.72 \pm 0.03$ |
| MultiHead | $1.18 \pm 0.02$ | $1.03 \pm 0.06$ | $1.15 \pm 0.16$ | $0.76 \pm 0.03$ | $0.64 \pm 0.02$ | $0.86 \pm 0.05$ | $0.74 \pm 0.06$ |
| MultipleModels-COM | $1.16 \pm 0.02$ | $0.89 \pm 0.03$ | $0.94 \pm 0.13$ | $0.73 \pm 0.02$ | $0.56 \pm 0.02$ | $0.68 \pm 0.07$ | $0.66 \pm 0.03$ |
| MultipleModels-ICT | $1.06 \pm 0.06$ | $1.09 \pm 0.04$ | $1.05 \pm 0.04$ | $0.75 \pm 0.04$ | $0.61 \pm 0.04$ | $0.75 \pm 0.04$ | $0.65 \pm 0.05$ |
| MultipleModels-IOM | $1.09 \pm 0.05$ | $1.03 \pm 0.06$ | $1.15 \pm 0.05$ | $0.67 \pm 0.04$ | $0.57 \pm 0.02$ | $0.73 \pm 0.06$ | $0.62 \pm 0.08$ |
| MultipleModels-RoMA | $1.02 \pm 0.03$ | $1.22 \pm 0.07$ | $0.93 \pm 0.13$ | $0.71 \pm 0.02$ | $0.59 \pm 0.01$ | $0.66 \pm 0.05$ | $0.59 \pm 0.05$ |
| MultipleModels-TriMentoring | $1.05 \pm 0.14$ | $0.82 \pm 0.11$ | $1.2 \pm 0.24$ | $0.74 \pm 0.02$ | $0.58 \pm 0.01$ | $0.78 \pm 0.07$ | $0.63 \pm 0.05$ |
| MultipleModels | $1.22 \pm 0.03$ | $1.07 \pm 0.12$ | $1.03 \pm 0.07$ | $0.82 \pm 0.03$ | $0.62 \pm 0.04$ | $0.83 \pm 0.08$ | $0.7 \pm 0.06$ |
| ParetoFlow | $0.88 \pm 0.05$ | $1.13 \pm 0.02$ | $1.05 \pm 0.02$ | $1.12 \pm 0.1$ | $1.11 \pm 0.07$ | $0.9 \pm 0.1$ | $0.68 \pm 0.02$ |
| **PGD-MOO + Data Pruning (Ours)** | $\mathbf{0.58 \pm 0.05}$ | $\mathbf{0.67 \pm 0.04}$ | $\mathbf{0.7 \pm 0.05}$ | $\mathbf{0.44 \pm 0.03}$ | $\mathbf{0.42 \pm 0.0}$ | $\mathbf{0.49 \pm 0.03}$ | $\mathbf{0.52 \pm 0.03}$ |
| **PGD-MOO (Ours)** | $0.56 \pm 0.02$ | $0.9 \pm 0.12$ | $0.64 \pm 0.04$ | $0.5 \pm 0.01$ | $0.43 \pm 0.02$ | $0.49 \pm 0.04$ | $0.58 \pm 0.04$ |

