# OpenReview forum: "Preference-Guided Diffusion for Multi-Objective Offline Optimization"
_NeurIPS.cc/2025/Conference — NeurIPS 2025 poster_

### Official Review · Reviewer_rK4P · 2025-06-01

**Clarity:** 3
**Significance:** 3
**Originality:** 3
**Rating:** 4
**Confidence:** 5

**Summary:**

This paper proposes a novel generative approach for offline multi-objective optimization using a classifier-guided diffusion model. The core idea is to guide the generation process using a preference model trained to predict whether one design dominates another. This preference model is not the same as the surrogate model generall used in offline MOO.  This guidance steers the diffusion model toward Pareto-optimal regions, including those beyond the training data.

**Questions:**

See Strengths And Weaknesses*

**Ethical Concerns:**

["NO or VERY MINOR ethics concerns only"]

**Final Justification:**

I’m leaning toward accepting.

**Limitations:**

See Strengths And Weaknesses*

**Quality:**

3

**Strengths And Weaknesses:**

Strengths: the paper handles an important problem and uses advanced techs to achieve impressive results.  I like the idea of using preference model to guide diffusion generation.

Weakness:
1. The paper misses an important citation: Kim M, Gu J, Yuan Y, et al. Offline Model-Based Optimization: Comprehensive Review[J]. arXiv preprint arXiv:2503.17286, 2025.

2. Figure 1 seems never mentioned in the text.

3. It looks like diversity contribution is dataset-dependent while the preference model is dataset-independent. How can u train a preference model using the diversity contribution or "equally dominant" is very rare?

4. I think the author may misunderstand some point: the preference model is trained on pairs from the offline dataset (time step 0) while used to guide designs across different time steps.

---

> ### Author Rebuttal · Authors · 2025-07-30
>
> Dear reviewer,
>
> Thanks a lot for your detailed and positive feedback. We are glad that you find our work addresses an important problem with impressive results. We address your concerns and questions below:
>
> - "_Missing Citation_": Thanks for bringing the Kim et al 2025 paper to our attention. While this paper directly does not address offline MOO, it does provide an overview of various approaches in the general offline setting. We will include the citation in the revision and discuss this aspect.
> - "_Reference to Figure 1_": Thanks for bringing this to our attention. We will include a reference to figure 1 in the main text.
> - “_How to train preference model when equally dominant is rare_”: That is a good question. We hypothesize that a sufficiently large dataset typically contains samples within each front, such that diversity can be computed. When the number of samples are low such that equally dominant points are rare, the preference model in this case would be trained more on the aspect of dominance rather than diversity. Given that the dataset is limited in this case, we argue that such a behaviour is expected and even preferable.
> - “_The preference model is trained on pairs from the offline dataset (time step 0) while used to guide designs across different time steps_”: The preference model is not just trained on the offline dataset at time step 0, but also on the noisy versions of the offline dataset, similar to what is done in classifier guidance for data modalities like images or text.
>
> We hope our response satisfactorily addressed all your concerns. If there are still any open questions or concerns, we are happy to answer them.

---

### Official Review · Reviewer_U1bb · 2025-06-25

**Clarity:** 2
**Significance:** 2
**Originality:** 2
**Rating:** 3
**Confidence:** 5

**Summary:**

The paper tackles the challenge of offline black-box multi-objective optimization (MOO), where new evaluations are infeasible, and the goal is to infer Pareto-optimal solutions beyond the observed dataset. To this end, the authors propose PGD-MOO, a novel inverse approach that combines diffusion models with preference-guided classifiers. Unlike forward methods that rely on training surrogate models for each objective, PGD-MOO leverages the generative capabilities of diffusion models to learn the mapping from objective values to input designs. Furthermore, a preference classifier model is trained to guide the denoising process toward the solution space. In addition to solution dominance, the classifier innovatively incorporates a diversity criterion to obtain more varied and sparsely distributed solutions. Experimental results demonstrate that PGD-MOO outperforms other inverse methods while maintaining strong generalization capabilities. It achieves solution quality comparable to forward methods, which may not be feasible in many real-world scenarios. Moreover, it exhibits significantly better diversity, enabling a more comprehensive approximation of the Pareto front.

**Questions:**

1. In practical optimization problems, there are often various constraints. How can the proposed method be extended to handle constrained multi-objective optimization problems and ensure that the generated solutions satisfy constraints?
2. The training process of diffusion models is often computationally intensive. Are there ways to optimize the training process to reduce computational costs and improve training efficiency, such as through model architecture design or training algorithm improvements?
3. As the number of objectives grows, Pareto dominance information tends to have a diminished impact. Why not consider incorporating hypervolume (HV) contribution or other metrics instead of relying solely on crowding distance when comparing two designs?
4. The experiments conducted thus far have focused on small-dimension datasets. How can the proposed method be effectively extended to tackle problems characterized by high-dimensionality and complex constraints?

**Ethical Concerns:**

["NO or VERY MINOR ethics concerns only"]

**Final Justification:**

I have carefully reviewed the author’s rebuttal and taken into account all the points raised. After thorough consideration, I believe that my original score remains appropriate and I will maintain it as is.

**Limitations:**

Yes

**Quality:**

3

**Strengths And Weaknesses:**

### Strengths
1. The paper pioneers a diffusion-based offline MOO method that integrates diversity optimization. It deftly replaces scalarized surrogates with a unified dominance/diversity classifier, thereby streamlining the overall architecture.
2. The method incorporates crowding distance (a concept borrowed from NSGA-II) into preference labeling, ensuring uniform and comprehensive coverage of the Pareto front.
3. Through evaluations on a diverse array of continuous offline multi-objective optimization tasks—spanning synthetic scenarios to real-world engineering problems—the proposed method consistently demonstrates superior performance. It excels in terms of both convergence to the Pareto front and the diversity of solutions.

### Weaknesses
1. The paper provides only a brief description of the diffusion model component. As the foundational element of the proposed method, the diffusion model's training process is not sufficiently detailed, particularly regarding how it is adapted to multi-objective optimization settings. The paper should elaborate on how multi-objective datasets are incorporated into the standard diffusion modeling pipeline and how the multi-objective nature of the problem is addressed during training.
2. In the experimental section, only one inverse approach is included for comparison, which may not be sufficient to fully demonstrate the advantages of the proposed method. It is recommended to include a diffusion-based baseline without classifier guidance as part of an ablation study to better highlight the contribution of the preference-guided mechanism.
3. The introduction section transitions rather abruptly into listing the contributions. Additionally, Section 3 offers relatively limited discussion of the proposed method, which appears disproportionate given the repeated exposition of the offline multi-objective optimization background earlier in the text. It is recommended to streamline the problem formulation to reduce redundancy and instead devote more space to elaborating on PGD-MOO, particularly the application and adaptation of the diffusion model.

---

> ### Author Rebuttal · Authors · 2025-07-30
>
> Dear reviewer,
>
> Thanks a lot for your detailed feedback. We are glad that you find that the strength of our work in excelling at both the diversity of solutions while also addressing the convergence to Pareto front. We address your concerns and questions below:
>
> - “_The paper provides only a brief description of the diffusion model component_” : We have explained the diffusion model in Sec 2.2 as well as the training details in Sec 5.1. The diffusion model is an unconditional one, i.e. it just models the distribution of the data. We incorporate multi-objective optimization entirely through our preference model, which is used for guidance. This is detailed in Sec 4. We wanted to keep the diffusion model explanation succinct in order to explain the main contribution of the paper, i.e. the preference model in more detail. But we are happy to add more details about diffusion models in the appendix for the camera-ready.
> - “_In the experimental section, only one inverse approach is included for comparison_”: ParetoFlow, which is another inverse approach, is the only approach which exists which directly addresses the problem of Offline MOO. If there are other inverse approaches which are relevant that we might have missed, please let us know and we are happy to compare them.
> - “_recommended to include a diffusion-based baseline without classifier guidance as part of an ablation study to better highlight the contribution of the preference-guided mechanism_”: Thanks a lot for the suggestion. In response to the reviewer 79Tz, we have provided results for ablation studies over the guidance weight $w$, including for the case of $w=0$, which corresponds to the simple diffusion based baseline without preference guidance. We find that having preference is essential to obtain designs close to the Pareto front. We will include these results in the camera-ready.
> - “_..introduction section transitions rather abruptly….It is recommended to streamline the problem formulation to reduce redundancy and instead devote more space to elaborating on PGD-MOO, particularly the application and adaptation of the diffusion model_”: Thanks for the suggestion. We have tried to structure the writing in such a way that the main contribution of the paper and the problem statement are laid out well. However, we will try to streamline a bit of the background section explain the diffusion model component in more detail in the revision.
> - “_In practical optimization problems, there are often various constraints. How can the proposed method be extended to handle constrained multi-objective optimization problems_”: That’s a great question. Note that input constraints—i.e., restricting inputs to a certain range—are already incorporated into our training procedure. If one instead wants to enforce constraints on the outputs (i.e., the objective values), this becomes a significantly harder problem, and to our knowledge, has not been addressed in the existing offline MOO literature. One intuitive way to approach this would be to train a classifier to distinguish between constraint-satisfying and constraint-violating solutions based on the training data, and then incorporate multiple classifier guidance during generation. However, this approach does not guarantee that the generated designs will always satisfy the constraints. We acknowledge that properly handling such output constraints requires careful treatment and is outside the scope of the current work.
> - “_The training process of diffusion models is often computationally intensive. Are there ways to optimize the training process to reduce computational costs_”: While training a diffusion model can be computationally expensive, our method benefits from general advancements in  faster training and sampling strategies proposed for diffusion models in the literature. For instance, one could use DDIM sampling [1] to speed up sampling. Other techniques used for faster training of diffusion models can also be used.
> - “_Why not consider incorporating hypervolume (HV) contribution or other metrics_”: Indeed, that is a very good point. While incorporating HV is desirable, it might be sensitive to scale due to the reference point. The actual value of HV can be different for different reference points. So for a practical problem where the reference point is harder to define, HV is harder to incorporate. However, if a reference point is well established, HV is definitely a reasonable alternative to define preference. In the response to reviewer 1hyQ, we have provided results on incorporating HV contribution as a way to measure diversity. We find that it is helpful, although the proposed _Crowding_ and _SubCrowding_ approaches are slightly better. We will add this discussion in the revision.
> - “_The experiments conducted thus far have focused on small-dimension datasets_": We have conducted experiments on settings where the dimensions are up to 30. The advantage of using a diffusion model is that it is shown to scale very well with the number of dimensions, as opposed to forward approaches based on surrogate models. We believe that a similar observation will hold true to this setting as well. Realistic high dimensional multiobjective datasets are also needed to thoroughly evaluate the model’s effectiveness for high-dimensional settings.
>
> We hope our response satisfactorily addressed all your concerns. If there are still any open questions or concerns, we are happy to answer them.
>
> ## References
>
> [1] Song, Jiaming, Chenlin Meng, and Stefano Ermon. "Denoising diffusion implicit models." arXiv preprint arXiv:2010.02502 (2020).

---

> > ### Comment · Reviewer_U1bb · 2025-08-05
> >
> > I appreciate that the authors have addressed some of my previous concerns. The paper claims to pioneer a diffusion-based method for offline multi-objective optimization (MOO), which is an interesting application. However, diffusion models have been widely explored in other fields, and their application to MOO seems like a natural extension rather than a groundbreaking innovation. The preference-guided classifier does add some novelty, but it is not entirely clear how much additional value it provides over existing methods that already incorporate diversity criteria.
> >
> > The experimental section includes only one inverse approach (ParetoFlow posted on arxiv) for comparison. While ParetoFlow is a relevant baseline, including more inverse approaches and a diffusion-based baseline without classifier guidance (as suggested in the rebuttal) would provide a more comprehensive evaluation and better demonstrate the advantages of the proposed method.
> >
> > The experiments are limited to small-dimension datasets, with the highest dimensionality being 30. While this is a reasonable starting point, real-world MOO problems often involve high-dimensional design spaces and complex constraints. The authors' assertion that realistic high-dimensional datasets are needed for thorough evaluation suggests that their method may not be fully ready for practical applications. It would be beneficial to see some preliminary results or discussions on how the method could be extended to higher-dimensional problems.
> >
> > The paper places significant emphasis on the diversity of solutions using the crowding distance metric. While crowding distance is a well-established metric, the authors acknowledge that incorporating hypervolume (HV) could be a better alternative in some cases. The limited exploration of HV and other metrics suggests that the paper's focus on diversity may be somewhat narrow and not fully representative of the broader goals of MOO. It would be valuable to see a more detailed comparison or discussion of different diversity metrics and their implications for the proposed method.
> >
> > Overall, the paper presents an interesting approach, but there is room for improvement in terms of novelty, comparative analysis, and presentation.

---

> ### Author Response · Authors · 2025-08-05
>
> Dear Reviewer,
>
> Thank you for your response. We would like to respectfully emphasize that the concerns raised in this round have already been addressed in the rebuttal. To avoid any misunderstanding, we briefly restate our responses below:
>
> - **"Existing methods that already incorporate diversity criteria":** To the best of our knowledge, no existing methods explicitly incorporate diversity criteria in the way we propose. If there is a specific method we have overlooked, we would be grateful if you could point it out, we would be happy to compare against it.
> - **"Only one inverse method compared":** As mentioned in our rebuttal, we are unaware of other inverse methods applicable in this context. Again, if there is a particular method we may have missed, we are open to including it.
> - **"Diffusion-based baseline without classifier guidance":** We have included these results in the rebuttal (see bullet point 3 in our rebuttal). If they do not sufficiently address your concern, we would appreciate further clarification.
> - **"Experiments are limited to small-dimension datasets":** We note that 30-dimensional problems are considered challenging in black-box optimization literature. We used the state of the art benchmarks in the literature.  Nonetheless, we are happy to evaluate our method on any specific higher-dimensional datasets you might suggest.
> - **"Limited exploration of alternate diversity measures":** As noted in our response to Reviewer 1hyQ, we included results using alternative diversity measures. We will add these results and a discussion to the final revision. Notably, we observed that crowding distance performed slightly better than HV.
>
> Kindly let us know if there are any concerns that still remain, especially regarding novelty or comparative analysis.

---

### Official Review · Reviewer_1hyQ · 2025-07-01

**Clarity:** 2
**Significance:** 2
**Originality:** 2
**Rating:** 4
**Confidence:** 4

**Summary:**

This paper presents Preference-Guided Diffusion for Multi-Objective Offline Optimization (PGD-MOO), which uses a preference-trained classifier to guide diffusion sampling in offline multi-objective optimization. A preference classifier is trained on pairwise comparisons in which the preferred design Pareto dominates its counterpart; when two designs are mutually equally dominating, the tie is resolved by favoring the one with a larger crowding distance, so the model learns to value diversity along the Pareto front. The authors demonstrate that PGD-MOO outperforms other baselines in DTLZ and ZDT tasks.

**Questions:**

1. Could you elaborate on why conditional guidance was chosen as the core mechanism for multi-objective offline optimization? In line 205, the paper mentions the goal of generating samples near the Pareto front, including those outside the training data distribution. However, Yuan et al [8] has suggested that conditional guidance can be difficult to tune effectively, and its generalization to out-of-distribution (OOD) regions tends to be limited. In contrast, approaches that fine-tune the diffusion model using proxy models such as DRaFT [9] and DDPO [10] have been shown to generate samples from desirable regions with strong performance. Finetuning methods are more robust in OOD area because finetuning moves the model distribution towards high-rewarding space. Moreover, recent approaches like GFlowNets [11] have been used to mitigate mode collapse in reinforcement learning, offering alternative sampling strategies with improved diversity. While [8][9][10][11] is not for MOO, in light of these alternatives, I am curious about the rationale behind choosing conditional guidance over diffusion finetuning methods.

2. A follow-up question related to the clarity of the methodology: could you confirm whether the preference model was trained using the Bradley-Terry model? This is suggested implicitly, but an explicit clarification would be very helpful.

[8] Yuan, Hui, et al. "Reward-directed conditional diffusion: Provable distribution estimation and reward improvement." Advances in Neural Information Processing Systems 36 (2023): 60599-60635.

[9] Clark, Kevin, et al. "Directly fine-tuning diffusion models on differentiable rewards." arXiv preprint arXiv:2309.17400 (2023).

[10] Black, Kevin, et al. "Training diffusion models with reinforcement learning." arXiv preprint arXiv:2305.13301 (2023).

[11] Liu, Zhen, et al. "Efficient Diversity-Preserving Diffusion Alignment via Gradient-Informed GFlowNets." arXiv preprint arXiv:2412.07775 (2024).

**Ethical Concerns:**

["NO or VERY MINOR ethics concerns only"]

**Final Justification:**

All of my concerns have been resolved by the authors through their rebuttal. I thank them for their efforts in providing thorough experimental results.

**Limitations:**

Yes.

**Paper Formatting Concerns:**

No concerns.

**Quality:**

2

**Strengths And Weaknesses:**

### Strengths
The paper is clearly structured and easy to follow. One of its most compelling aspects is the idea of overweighting more diverse samples over concentrated ones, thereby encouraging the classifier to favor diversity. To the best of my knowledge, this is a novel approach, and it appears to improve the ∆-spread metric effectively. In the synthetic function benchmarks, PGD-MOO shows that a preference-based classifier outperforms the surrogate model approach, as well as multi-head and multi-model baselines.

### Weakness
- **Narrow experimental coverage.** PGD-MOO is tested on two synthetic tasks (DTLZ, ZDT) and the RE engineering tasks, whereas Xue et al. [1] include a broader set that includes MO-Swimmer, MO-Hopper, MO-Portfolio, and Molecule tasks, characterized by continuously represented data. It would be better to add more continuous representation tasks to the experiment section for more solid demonstrations.

- **PGD-MOO underperforms ParetoFlow** On the RE suite, PGD-MOO underperforms ParetoFlow [2] despite using the same guidance mechanism—the main difference being the classifier. This result suggests that a preference-based classifier may not outperform a surrogate regressor in practice.

- **Limited methodological novelty.** Classifier-guided diffusion sampling was first introduced by Dhariwal et al. [3] and later extended to classifier-free guidance by Ho et al. [4]; these two variants are fundamentally the same in that both condition the reverse process at inference time. For offline optimization, Krishnamoorthy et al. [5] already explored classifier-free guidance for offline model-based optimization. Even if one considers conditioned generation to be a universal technique, the use of preference classifiers is already a well-established idea in RLHF for language models [6]. As such, the novelty of this work remains incremental.

- **Restricted to continuous design spaces.** Because PGD-MOO inherits the continuous DDPM backbone [7], it cannot natively handle discrete variables, whereas ParetoFlow [2] works for both discrete and continuous problems. Adopting a discrete-friendly diffusion variant (or a data-type–agnostic flow) would make the approach more generally useful.

- **Clarity.** The explanation of the training method for the preference model in Section 4.1 could benefit from greater clarity. It seems to be based on the Bradley-Terry model, but this is not made explicit in the text. Providing a more detailed and transparent description would help readers better understand the underlying methodology.

- **Limited ablation about diversity contribution.** The paper provides limited ablation on the *diversity contribution* to the selection of winning samples. To better understand the effectiveness of isolating diversity as a design choice, it would be valuable for the authors to include an ablation study specifically examining its impact on performance.

[1] Xue, Ke, et al. "Offline multi-objective optimization." arXiv preprint arXiv:2406.03722 (2024).

[2] Yuan, Ye, et al. "ParetoFlow: Guided Flows in Multi-Objective Optimization." arXiv preprint arXiv:2412.03718 (2024).

[3] Dhariwal, Prafulla, and Alexander Nichol. "Diffusion models beat gans on image synthesis." Advances in neural information processing systems 34 (2021): 8780-8794.

[4] Ho, Jonathan, and Tim Salimans. "Classifier-free diffusion guidance." arXiv preprint arXiv:2207.12598 (2022).

[5] Krishnamoorthy, Siddarth, Satvik Mehul Mashkaria, and Aditya Grover. "Diffusion models for black-box optimization." International Conference on Machine Learning. PMLR, 2023.

[6] Ouyang, Long, et al. "Training language models to follow instructions with human feedback." Advances in neural information processing systems 35 (2022): 27730-27744.

[7] Ho, Jonathan, Ajay Jain, and Pieter Abbeel. "Denoising diffusion probabilistic models." Advances in neural information processing systems 33 (2020): 6840-6851.

---

> ### Author Rebuttal · Authors · 2025-07-30
>
> Dear Reviewer,
>
> Thanks a lot for your detailed feedback. We are glad that you find our work structured and easy to follow as well as our approach novel in terms of incorporating diversity for offline MOO.
> Below we address the questions and concerns you have raised:
>
> - “_Narrow experimental coverage_”: We would like to emphasize that DTLZ, ZDT and RE contain multiple datasets within each task, with various objectives and dimensions. The total number of datasets is 17. In this regard, we believe testing on these tasks already provides a good representation of various approaches, including our method. However, we are happy to include additional experiments on representative tasks like Multi-Objective  NAS and Multi-Objective RL. Given the limited space for the rebuttal, we will make these results available in the revision. Note that the provided code, which will be made available on acceptance, has the option to run it on all datasets and tasks that are part of the benchmark Xue et al 2024 paper.
> - “_PGD-MOO underperforms ParetoFlow On the RE suite_”: We find that ParetoFlow is slightly better in terms of hypervolume than our approach when the number of objectives is 4 or greater, but our method is competitive or better otherwise. More importantly, our approach results in much more diverse solutions than ParetoFlow, an important aspect in MOO that has not been given enough attention so far. In this regard, we argue that our method is still better in practice when considering both diversity and hypervolume simultaneously. We have reported all findings transparently to highlight both the strengths and weaknesses of all methods, and we hope this balanced presentation will not be held against our paper.
> - “_Limited methodological novelty_“: Our method is novel across two different aspects - 1. A classifier based on preference defined by Pareto dominance and 2. As you have also acknowledged, ensuring diversity of resulting samples by explicitly training the classifier using the preference samples. While conditional diffusion and the use of preference in RLHF has been done before for other problems, we believe that the aspect of preference from a Pareto viewpoint while addressing the aspect of diversity is entirely novel for MOO.
> - “_Restricted to continuous design spaces_“: We note that the ParetoFlow method cannot be natively adapted to combinatorial problems. However, they do address the problem of discrete (categorical) cases by treating the discrete classes as continuous (through the logits of one hot representation). Since this essentially renders the approach as continuous, we can adopt the same approach for our method for discrete cases. Multi-Objective Neural Architecture Search benchmark provides such a case. As mentioned before, we will include this result in the revision.
> - “_Clarity,......,The explanation of the training method for the preference…. Is it following the Bradley-Terry Model_”: Thanks for bringing this point. Indeed, our preference model is based on Bradley-Terry. We train the preference model as just a binary classifier over preference pairs (which are designs with well defined dominance relations). During sampling, we use this trained preference model to guide the diffusion model using the logits of the class which indicates the first input dominates the other. This logit could be interpreted as the “reward” to the diffusion model for preferring the first input over the second, which corresponds to the Bradley-Terry setup. We will make this aspect more explicit in the revision.
> - “_Limited ablation about diversity contribution_" : Thanks for the suggestion. We perform ablation experiments on diversity contribution. Due to the space and limitation, we are including only a subset of representative results below, but we will include the full set of results in the appendix in the camera-ready. Apart from the Crowding and SubCrowding approaches which are explained in the paper, we also include results for two different diversity criteria - _No Div. Criteria_ and _HV_. _No Div. Criteria_ is the approach without any explicit diversity contribution and HV is the approach with diversity measured as the hypervolume improvement due to any sample in the dataset.  We find that incorporating diversity criteria for preference generally results in better diversity of the resulting solutions. In addition, Crowding and SubCrowding gives better diversity than using hypervolume improvement. We provide detailed results below (all results are evaluated on 256 solutions with 5 random seeds):
>
> ## ZDT Tasks -Spread (Lower is better)
>
> | Diversity Criteria   | ZDT1          | ZDT2          | ZDT3          | ZDT4          | ZDT6          |
> |:---------------------|:--------------|:--------------|:--------------|:--------------|:--------------|
> | No Div. Criteria        | 0.72$\pm$0.04 | 0.60$\pm$0.03 | 0.66$\pm$0.03 | 0.65$\pm$0.00 | 0.84$\pm$0.01 |
> | Crowding             | 0.63$\pm$0.09 | 0.58$\pm$0.01 | 0.60$\pm$0.02 | 0.60$\pm$0.02 | 0.83$\pm$0.04 |
> | HV                   | 0.64$\pm$0.06 | 0.61$\pm$0.03 | 0.60$\pm$0.04 | 0.60$\pm$0.02 | 0.86$\pm$0.05 |
> | SubCrowding          | 0.70$\pm$0.02 | 0.90$\pm$0.04 | 0.66$\pm$0.03 | 0.60$\pm$0.04 | 0.77$\pm$0.06 |
>
> ## RE Tasks - Spread (Lower is better)
>
> | Diversity Criteria   | RE21          | RE24          | RE25          | RE35          | RE41          |
> |:---------------------|:--------------|:--------------|:--------------|:--------------|:--------------|
> | No Div. Criteria        | 0.62$\pm$0.01 | 1.16$\pm$0.06 | 1.39$\pm$0.06 | 0.90$\pm$0.02 | 0.50$\pm$0.04 |
> | Crowding             | 0.61$\pm$0.02 | 1.20$\pm$0.05 | 1.19$\pm$0.08 | 1.00$\pm$0.12 | 0.47$\pm$0.04 |
> | HV                   | 0.62$\pm$0.02 | 1.16$\pm$0.09 | 1.14$\pm$0.09 | 0.99$\pm$0.08 | 0.47$\pm$0.03 |
> | SubCrowding          | 0.60$\pm$0.01 | 1.17$\pm$0.05 | 1.12$\pm$0.06 | 0.70$\pm$0.03 | 0.43$\pm$0.02 |
>
> - “_I am curious about the rationale behind choosing conditional guidance over diffusion finetuning methods_”: We choose conditional guidance as it is a relatively simple and straightforward method to generate samples beyond the model training distribution. We notice that for the datasets and benchmarks that we have tested these models on, they do in fact generate samples from beyond the training data, close to the Pareto front. However, diffusion finetuning methods are definitely an interesting alternative. We note that even when diffusion finetuning methods are used, conditional guidance usually has complementary strengths that can be used to further enhance the quality of designs.
> - “_It appears that the references cited in the main text (e.g., [1]) are not listed in the References section_”: Xue et al. 2024 has been listed in line 459 in references. Kindly let us know if we misunderstood something regarding this comment.
>
> We hope our response satisfactorily addressed all your concerns. If there are still any open questions or concerns, we are happy to answer them.

---

> > ### Comment · Reviewer_1hyQ · 2025-08-02
> >
> > Thank you for taking the time to provide a thorough rebuttal to my review. I appreciate the detailed clarifications, particularly regarding paper clarity and the ablation results. These have addressed many of the concerns I initially raised, and I thank the authors for their efforts in this regard.
> >
> > However, a few important concerns remain unresolved. In particular, my earlier point about the limitations of classifier-guided conditional guidance and the fact that PGD-MOO underperforms ParetoFlow on the RE benchmark tasks still stand. This leads me to the following conclusion: among the main differences between ParetoFlow and PGD-MOO, the most significant is the training method used for the classifier, whether using the Bradley-Terry model or a standard regression model. Given that the Bradley-Terry-based approach performs worse on the RE tasks, this raises critical concerns about the effectiveness of preference-based learning.
> >
> > Additionally, as you noted in the rebuttal, PGD-MOO has results for DTLZ and ZDT, and you have indicated that further results (including NAS and Multi-Objective RL tasks) will be added in the revised version. However, I have not yet been able to verify these results. If experimental results on discrete domains such as NAS and Multi-Objective RL tasks could be shared during the discussion period, it would be very helpful.
> >
> > To summarize my remaining concerns:
> >
> > 1. PGD-MOO’s underperformance on the RE tasks relative to ParetoFlow raises significant doubts about the effectiveness of the Bradley-Terry–based guidance method.
> > 2. I have requested experimental results for discrete representations (e.g., NAS) and Multi-Objective RL tasks, but these results have not yet been provided.
> >
> > If you can provide a stronger response to these two concerns, I will actively consider raising my score toward the acceptance threshold.
> >
> > Thank you.

---

> > > ### Author Response · Authors · 2025-08-08
> > >
> > > Dear Reviewer,
> > >
> > > Thank you for your response. We’re glad to hear that our rebuttal addressed most of your concerns. To fully address the remaining points, we ran additional experiments on the Multi-Objective Neural Architecture Search (MO-NAS) tasks—18 in total, spanning the C10MOP and IN1KMOP datasets—on the MultiHead baseline, the ParetoFlow baseline, and our approach, evaluated on the latest datasets and benchmarks from Xue et al. (2024). Across all settings, our method consistently outperforms all baselines, including in high-objective cases such as C10MOP6 (6 objectives) and C10MOP7 (8 objectives). These gains are particularly noteworthy given the discrete nature of the data/design space, which we convert to one-hot logits and train in the continuous space, as in ParetoFlow. Detailed results on hypervolume are included below (all results are evaluated on 256 solutions across 5 seeds, with **bold** denoting the best result and *italic* denoting the second best):
> > >
> > > ## C10MOP Datasets
> > >
> > > | Approach | C10MOP1 | C10MOP2 | C10MOP3 | C10MOP4 | C10MOP5 | C10MOP6 | C10MOP7 | C10MOP8 | C10MOP9 |
> > > |----------|----------|----------|----------|----------|----------|----------|----------|----------|----------|
> > > | MultiHead | 1.4559 ± 0.0227 | 1.2746 ± 0.0588 | 10.2391 ± 0.2114 | 21.2637 ± 1.2549 | 36.0405 ± 3.3757 | 98.8094 ± 0.6955 | 340.8008 ± 60.0297 | 5.0691 ± 0.1096 | 11.4813 ± 0.5663 |
> > > | MultiHead-GradNorm | 1.4309 ± 0.0155 | 1.2636 ± 0.0572 | 10.0193 ± 0.2474 | 20.7159 ± 1.0982 | 30.5653 ± 5.2836 | 77.9574 ± 11.9034 | 306.7220 ± 39.5226 | 5.3211 ± 0.1007 | 12.9097 ± 0.5352 |
> > > | MultiHead-PcGrad | *1.4601 ± 0.0102* | 1.3023 ± 0.0787 | 10.2801 ± 0.2455 | 21.4537 ± 0.6093 | 38.9828 ± 5.0494 | 88.2995 ± 9.7358 | 360.9641 ± 30.9560 | 4.9822 ± 0.0631 | 12.6775 ± 0.3565 |
> > > | ParetoFlow | 1.4456 ± 0.0171 | 1.3437 ± 0.0164 | 10.6091 ± 0.2478 | 21.6532 ± 0.6013 | 48.6245 ± 0.8522 | 104.8361 ± 1.9998 | 463.2916 ± 21.6113 | 5.2958 ± 0.0921 | *14.3905 ± 0.2319* |
> > > | **PGD-MOO + Pruning (Ours)** | **1.4817 ± 0.0021** | **1.3817 ± 0.0138** | *11.1524 ± 0.0165* | *23.8769 ± 0.1086* | *49.8578 ± 0.0266* | *106.6697 ± 0.9115* | **500.4782 ± 1.7625** | **5.5659 ± 0.0156** | **14.4528 ± 0.1726** |
> > > | **PGD-MOO (Ours)** | 1.4511 ± 0.0141 | *1.3598 ± 0.0105* | **11.1826 ± 0.0148** | **23.9233 ± 0.0821** | **49.8683 ± 0.0658** | **107.2650 ± 0.6335** | *499.1712 ± 1.3745* | *5.4807 ± 0.0363* | 14.0768 ± 0.2056 |
> > >
> > > ## IN1KMOP Datasets
> > >
> > > | Approach | IN1KMOP1 | IN1KMOP2 | IN1KMOP3 | IN1KMOP4 | IN1KMOP5 | IN1KMOP6 | IN1KMOP7 | IN1KMOP8 | IN1KMOP9 |
> > > |----------|----------|----------|----------|----------|----------|----------|----------|----------|----------|
> > > | MultiHead | 5.6206 ± 0.0161 | 5.8983 ± 0.1535 | 14.2471 ± 0.1133 | 4.1108 ± 0.1259 | 4.2844 ± 0.0877 | 10.3601 ± 0.0467 | 4.8340 ± 0.0840 | 9.2124 ± 0.1680 | 10.2314 ± 0.9201 |
> > > | MultiHead-GradNorm | 5.1362 ± 0.5426 | 6.2062 ± 0.0950 | 14.0209 ± 0.1754 | 3.1755 ± 0.0474 | 4.2984 ± 0.6297 | 8.1613 ± 0.7969 | 4.6107 ± 0.2090 | 10.1800 ± 0.6765 | 11.4468 ± 1.1444 |
> > > | MultiHead-PcGrad | 5.5797 ± 0.0810 | 6.0273 ± 0.3497 | 14.2122 ± 0.0955 | 4.2014 ± 0.0729 | 4.2996 ± 0.0449 | 10.8542 ± 0.1305 | 4.8260 ± 0.3493 | 8.7804 ± 0.2875 | 10.2910 ± 0.0455 |
> > > | ParetoFlow | 5.3459 ± 0.1106 | 5.9646 ± 0.1658 | 14.3332 ± 0.2594 | 4.3161 ± 0.0514 | 4.6240 ± 0.0506 | 11.6062 ± 0.0935 | 4.8707 ± 0.0423 | **11.1389 ± 0.1251** | **14.4814 ± 0.1094** |
> > > | **PGD-MOO + Pruning (Ours)** | *5.7570 ± 0.0664* | **6.4504 ± 0.0201** | *14.7568 ± 0.1310* | *4.3928 ± 0.0296* | *4.6531 ± 0.0426* | *11.7936 ± 0.1010* | **5.3022 ± 0.0705** | 10.6270 ± 0.1074 | *13.9399 ± 0.2251* |
> > > | **PGD-MOO (Ours)** | **5.7638 ± 0.0489** | *6.4305 ± 0.0232* | **14.8635 ± 0.1562** | **4.4924 ± 0.0105** | **4.7465 ± 0.0146** | **11.9365 ± 0.0626** | *5.1797 ± 0.0780* | *10.7654 ± 0.1653* | 13.6189 ± 0.3393 |
> > >
> > > With these 18 new discrete datasets—on top of the 17 datasets already reported in the paper—we believe your remaining concerns are directly addressed. We will include these results in the revision. While time and space limitations during the rebuttal phase prevent us from running the full MORL experiments right now, we will include them in the revision as well. As noted earlier, we will also release the code, which supports all of these tasks.
> > >
> > > Regarding the "*effectiveness of the Bradley-Terry–based guidance method*”: The additional results above clearly demonstrate that our approach not only matches but outperforms ParetoFlow, even in challenging settings. It’s worth emphasizing that offline MOO is still a young and rapidly evolving field, and diffusion/flow-based approaches are relatively new compared to search-based ones. Our work introduces a novel alternative guidance strategy, expands the discussion on inverse approaches, and highlights the importance of diversity. We believe that our work is a timely and meaningful contribution to the community in this regard.
> > >
> > > Please let us know if there are still remaining concerns/ questions.

---

### Official Review · Reviewer_79Tz · 2025-07-03

**Clarity:** 3
**Significance:** 3
**Originality:** 3
**Rating:** 5
**Confidence:** 4

**Summary:**

This paper proposes a diffusion-based generative method for offline MOO, which aims to identify the Pareto-optimal solutions only given the offline dataset $\mathcal{D}= \\{ ( \mathbf{x}_i, \mathbf{y}_i ) \\} _{i=1}^{N}$. The proposed method first trains a pairwise preference classifier based on NSGA-II selection [1] (integrating both dominance and solution diversity) to determine which design is better, then trains an unconditioned diffusion model. After that, it uses the classifier to guide the denoising procedure. Experimental results demonstrate the effectiveness of the proposed method.

**Questions:**

How do you set up the noise schedules $\beta_t$ in your experiments?

**Ethical Concerns:**

["NO or VERY MINOR ethics concerns only"]

**Final Justification:**

This paper proposes a method based on diffusion model and classifier-based preference guider for offline multi-objective optimization. The method incorporates training unconditional diffusion model $p(x)$ and sampling with classifier guidance, where the classifier is based on NSGA-II selection. It is different from recent offline MOO work like ParetoFlow. This method is direct and intuitive, and the experimental results are competitive.

Other reviewers mainly have concerns on: (1) missing related discussion of diffusion models; (2) limited experiments on Off-MOO-Bench [1]. For (1), the authors have added necessary discussion. For (2), there are so many tasks in [1] where the evaluations of some tasks are expensive, and adding all tasks during rebuttal is difficult. The authors have promised to add more tasks in the revised version. Besides, for optimization tasks that have constraints raised by other reviewers, studies in offline black-box optimization rarely consider this situation, and the authors have discussed potential solutions.

Overall, I believe this paper would be of high quality after revision. Thus I've raised my score.

[1] Offline multi-objective optimization. ICML 2024.

**Limitations:**

Limited experimental studies:

- Effectiveness of the classification-based preference guider is not examined. For example, I suggest compare it with weighted sum regression-based predictor [4] or ranking-based preference guider [5] (use ranking loss to train the model).
- I cannot understand the benefits of the inverse approaches in offline MOO based on the experiments. Upon the same preference classifier, I suggest comparing your diffusion-based method to searching inside the classifier by replacing the selection operator of NSGA-II with the model’s output.
- Ablation of the guidance weight $w$ is needed, as in [6].

I understand that conducting additional experimental evaluations in all tasks of Off-MOO-Bench [2] is time-consuming during the rebuttal period. I suggest that you compare those in some representative tasks. I will be glad to increase the score once you address my concerns.

## References

[1] A fast and elitist multiobjective genetic algorithm: NSGA-II. IEEE TEvC, 2002.

[2] Offline multi-objective optimization. ICML 2024.

[3] Robust Guided Diffusion for Offline Black-Box Optimization. TMLR 2024.

[4] ParetoFlow: Guided Flows in Multi-Objective Optimization. ICLR 2025.

[5] Offline model-based optimization by learning to rank. ICLR 2025.

[6] Diffusion models for black-box optimization. ICML 2023.

**Quality:**

2

**Strengths And Weaknesses:**

## Strengths

- This proposed method is clear and well motivated.
- The code implementation is clean and well constructed.

## Weaknesses

- In line 253, you mentioned that “This approach does not require training surrogate models, thus providing a simple alternative approach to offline MOO”. As in [2], the end-to-end surrogate model is prediction-based MLP regressor. Don’t you need to train an MLP-based preference classification-based surrogate model too? Based on my comprehension, this discussion is not appropriate.
- Based on my comprehension, your approach is somehow similar to RGD [3] in offline single-objective optimization. Relevant discussion on both this work and the difference between your framework and RGD is needed.

---

> ### Author Rebuttal · Authors · 2025-07-30
>
> Dear Reviewer,
>
> Thanks a lot for your thoughtful and detailed feedback. We are glad that you found our work clear and well motivated with a clean constructed code.
> We answer the concerns and questions you have raised below:
>
> - “_Don’t you need to train an MLP-based preference classification-based surrogate model too?_”: In line 253, we refer to surrogate models as they are commonly used in the literature, namely, models that predict the output $y$ for any input $\mathbf{x}$. In multi-objective optimization (MOO) settings, forward approaches typically train a separate surrogate model for each objective, and these are then used to search over a pre-defined design space. This is in contrast to our method, where the classifier is used to guide the diffusion-based generation process, rather than search within a fixed design space. This distinction is important: in inverse approaches like ours and Pareto Flow, no explicit search is required. Instead, designs are generated directly by the diffusion model, guided by learned preferences or classifiers, without needing to enumerate or evaluate points in a pre-set space.
> - “_your approach is somehow similar to RGD [3] in offline single-objective optimization_”: Thanks for bringing this paper to our attention. As you mention, RGD focuses mainly on offline single-objective setting, whereas the goal of this work is to explicitly address the harder problem of multi-objective setting, where the notions of Pareto Front and Pareto dominance arise. In addition, while RGD uses guided diffusion like the ParetoFlow approach, the guidance mechanism is different from ours, wherein we incorporate Pareto dominance relations for the guidance classifier. We will include RGD in related work and discuss these differences.
> - “_How do you set up the noise schedules $\beta_t$ in your experiments?_”: Following the DDPM paper (Ho et al. 2020), we set the noise schedule $\beta_t$ such that it grows linearly from 1e-4 to 0.02 with time $t$. We will add this detail in the experiment section.
> - “_Effectiveness of the classification-based preference guider is not examined. For example, I suggest compare it with weighted sum regression-based predictor [4] or ranking-based preference guider [5] (use ranking loss to train the model)_”: Thanks for the suggestion. [4] is the Pareto Flow approach, which is an inverse model similar to ours, and we compare to it extensively in our work. We find that our approach performs competitively across many datasets as compared to ParetoFlow. [5] mainly deals with single objective settings, and thus a direct comparison is not feasible since the actual outputs (a single best design vs the pareto set) and the corresponding metrics (hypervolume vs $y$ spread) would be very different.
>
> - “_cannot understand the benefits of the inverse approaches in offline MOO based on the experiments. Upon the same preference classifier, I suggest comparing your diffusion-based method to searching inside the classifier by replacing the selection operator of NSGA-II with the model’s output_”: Thank you for your suggestion. As explained in our introduction, there are several important settings—such as drug discovery—where generating entirely new designs is essential, and relying on a pre-defined search space is not always feasible. In such cases, inverse approaches become crucial and are often the only viable strategy. Forward approaches that depend on search procedures and the output of a surrogate model are not practical in these contexts. We do, however, compare against several search-based methods in our experiments for benchmarks where this is a feasible approach, which aligns with your recommendation.
> - “_Ablation of the guidance weight $w$ is needed_": Thanks for the suggestion. We perform ablation experiments on the guidance weight $w$. Due to the space limitation, we are including only a subset of representative results below, but we will include the full set of results in the appendix in the camera-ready. We find that the guidance weight has an effect on both the hypervolume and diversity. With $w=0$ (no preference guidance), the results are lower, as expected. Increasing the guidance weight generally results in better hypervolume at the slight cost of diversity. We provide detailed results below (all results are evaluated on 256 solutions with 5 random seeds):
>
>
> ### ZDT TASKS - Hypervolume Metric
>
> | Guidance | ZDT1 | ZDT2 | ZDT3 | ZDT4 | ZDT6 |
> | :--- | :---: | :---: | :---: | :---: | :---: |
> | $w=0$ | 4.53±0.04 | 5.20±0.03 | 5.57±0.09 | 5.08±0.04 | 4.60±0.12 |
> | $w=5$ | 4.53±0.04 | 5.31±0.03 | 5.55±0.03 | 5.06±0.07 | 4.81±0.02 |
> | $w=10$ | 4.55±0.04 | 5.39±0.05 | 5.55±0.06 | 4.97±0.07 | 4.82±0.00 |
> | $w=20$ | 4.53±0.13 | 5.52±0.02 | 5.60±0.10 | 5.00±0.07 | 4.82±0.01 |
> | $w=50$ | 4.51±0.18 | 5.62±0.04 | 5.55±0.04 | 5.04±0.04 | 4.63±0.33 |
>
> ### ZDT TASKS - Spread Metric (Lower is better)
>
> | Guidance | ZDT1 | ZDT2 | ZDT3 | ZDT4 | ZDT6 |
> | :--- | :---: | :---: | :---: | :---: | :---: |
> | $w=0$ | 0.64±0.04 | 0.62±0.01 | 0.62±0.04 | 0.61±0.02 | 0.77±0.05 |
> | $w=5$ | 0.63±0.03 | 0.66±0.03 | 0.62±0.02 | 0.61±0.01 | 0.81±0.03 |
> | $w=10$ | 0.63±0.09 | 0.58±0.01 | 0.60±0.02 | 0.60±0.02 | 0.83±0.04 |
> | $w=20$ | 0.63±0.02 | 0.58±0.02 | 0.60±0.02 | 0.58±0.02 | 0.90±0.06 |
> | $w=50$ | 0.60±0.03 | 0.60±0.02 | 0.62±0.03 | 0.62±0.01 | 0.91±0.04 |
>
> ### RE TASKS - Hypervolume
>
> | Guidance Weight | RE21 | RE24 | RE25 | RE35 | RE41 |
> | :--- | :---: | :---: | :---: | :---: | :---: |
> | $w=0$ | 4.38±0.07 | 4.84±0.00 | 4.84±0.00 | 10.42±0.06 | 18.98±0.25 |
> | $w=5$ | 4.43±0.05 | 4.84±0.00 | 4.84±0.00 | 10.37±0.06 | 19.00±0.26 |
> | $w=10$ | 4.46±0.04 | 4.83±0.00 | 4.84±0.00 | 10.27±0.11 | 19.17±0.30 |
> | $w=20$ | 4.45±0.05 | 4.84±0.00 | 4.84±0.00 | 10.16±0.15 | 18.40±0.84 |
> | $w=50$ | 4.44±0.05 | 4.83±0.00 | 4.84±0.00 | 9.55±0.54 | 17.67±0.96 |
>
> ### RE TASKS - Spread (Lower is better)
>
> | Guidance Weight | RE21 | RE24 | RE25 | RE35 | RE41 |
> | :--- | :---: | :---: | :---: | :---: | :---: |
> | $w=0$ | 0.60±0.03 | 1.15±0.07 | 1.05±0.05 | 0.70±0.03 | 0.42±0.02 |
> | $w=5$ | 0.61±0.04 | 1.20±0.09 | 1.08±0.05 | 0.81±0.12 | 0.45±0.02 |
> | $w=10$ | 0.61±0.02 | 1.20±0.05 | 1.19±0.08 | 1.00±0.12 | 0.47±0.04 |
> | $w=20$ | 0.61±0.03 | 1.18±0.03 | 1.21±0.06 | 0.83±0.11 | 0.49±0.04 |
> | $w=50$ | 0.63±0.02 | 1.17±0.08 | 1.21±0.06 | 0.82±0.09 | 0.62±0.19 |
>
>
> - “_Additional experiments on representative tasks_”: Thanks for the suggestion. We extensively evaluate 17 different datasets across three different sets of tasks DTLZ, ZDT and RE. However, we are happy to include additional experiments on representative tasks like Multi-Objective  NAS and Multi-Objective RL. Given the limited space for the rebuttal, we will make these results available in the revision. Note that the provided code, which will be made available on acceptance, has the option to run it on all datasets and tasks that are part of the benchmark Xue et al 2024 paper.
>
> We hope our response satisfactorily addresses all your concerns. If there are still any open questions or concerns, we are happy to answer them.
>
> References:
>
> Ho, J., Jain, A., & Abbeel, P. (2020). Denoising Diffusion Probabilistic Models. Advances in Neural Information Processing Systems (NeurIPS).

---

> > ### Comment · Reviewer_79Tz · 2025-08-02
> >
> > Thanks for your prompt response. The clarification of the differences between your work, RGD, and ParetoFlow is clear and convincing, where your sampling based on Pareto dominance and crowding distance aligns well with MOO, reflecting more precise Pareto-optimal solutions. From my perspective, the differences in sampling shares similar differences between NSGA-II (yours) and MOEA/D (scalarization-based methods like ParetoFlow). I suggest making a more detailed discussion of PGD and ParetoFlow. I can understand your contribution based on your discussion, which utilizes proxy-guided generative models and integrates the properties of MOO. After reading other reviewers' comment, I decide to increase my score from 3 to 5 and lean towards accepting this paper!
> >
> > Additionally, I have noticed that the Off-MOO-Bench repo has incorporated ParetoFlow (https://github.com/lamda-bbo/offline-moo/tree/main/off_moo_baselines/paretoflow). I would suggest you contributing your implementation code to this repo  to further enhance the follow-up studies :-)

---

> > > ### Author Response · Authors · 2025-08-09
> > >
> > > Dear reviewer,
> > >
> > > Thanks a lot for your helpful feedback. We are glad that we were able to address all your concerns and appreciate the increasing in the rating. Thanks for the excellent suggestion on contributing our code to the official Off-MOO Bench repo to further enhance the follow-up studies, we will do so on acceptance of this paper.

---

### Note · Authors · 2025-08-15

Dear Area Chairs and Reviewers,

Thanks a lot for the positive feedback and for suggesting improvements for the submitted work. As the rebuttal and discussion period is coming to an end, we would  like to summarize the discussion and highlight additional experiments performed to address the concerns of the reviewers:

- **Experiments on the new task**: We have provided results on the MO-NAS discrete task (18 datasets in total) and find that our approach performs better than ParetoFlow and search-based baselines. This is in addition to the 17 datasets that are already evaluated on continuous domains.
- **Ablation experiments**: We have provided ablation results for the following cases: analyzing the diffusion guidance weight $w$ and different variations of incorporating diversity. In addition, with diffusion guidance weight $w=0$, it also includes the case of an unconditional diffusion model without guidance. We find that preference guidance is helpful, and that increasing guidance weight increases hypervolume at the slight cost of diversity. In addition, we find that the proposed diversity measure using the Crowding distance usually gives better diversity of the resulting solutions.

We will incorporate these additional experiments in the revision.

We believe we have addressed all the concerns of the reviewers with relevant experiments and clarifications. We hope that this is taken into account to enable a positive assessment of our paper.

---

### Decision · Program_Chairs · 2025-09-17

**Decision:**

Accept (poster)

**Comment:**

This work proposes PGD-MOO, a preference-guided diffusion method for tackling multi-objective offline optimization problems. The key idea of PGD-MOO is to build a diffusion model with preference-based classifier guidance to generate Pareto optimal solutions based on offline dataset. Inspired by the idea of the seminal multi-objective optimization algorithm NSGA-II, the guidance classifier is trained to predict the domination relation between two different solutions, which can then be used to direct the diffusion model to generate Pareto optimal solutions (the ones that dominate other non Pareto optimal solutions). In addition, a diversity-aware preference guidance approach using crowding distance in NSGA-II is proposed to further encourage the model to generate well-distributed solutions on the Pareto front. Experimental results show PGD-MOO can achieve promising performance on the ZDT/DTLZ, RE, and MO-NAS problems (provided in the rebuttal) under the offline optimization setting.

The reviewers find this paper clearly structured and easy to follow, the proposed PGD-MOO method clear and well-motivated, the diversity-aware approach compelling and can ensure uniform coverage of the Pareto front, and the code implementation clean and well-constructed. Some concerns have been raised on clarifications, relations to existing works, missing citation, methodological novelty and extension, experimental coverage and performance. Most of these concerns have been properly addressed during the rebuttal.

Finally, this work has mixed ratings, where one reviewer votes to clearly accept this work, 2 reviewers lean toward weak acceptance, and 1 reviewer recommends weak rejection. In the final justification, the most positive reviewer 79Tz believes this paper would be of high quality after revision. The original most negative reviewer 1hyQ believes all of his/her concerns have been resolved and hence raised the rating to weak acceptance. The remaining negative reviewer U1bb did not respond to the author's further response.

I read the paper myself and believe that offline multi-objective optimization is an important problem, and this work is a timely and valuable contribution to this research direction. I agree with the reviewers that the proposed PGD-MOO method with a diversity-aware approach is well-motivated and compelling. In addition, I believe the authors' rebuttal has properly addressed the concerns raised by reviewer U1bb, and the new results on 18 practical MO-NAS problems provided in the rebuttal are also highly appreciated. Therefore, I recommend acceptance of this work.

Please make sure all the discussion and revisions are carefully incorporated into the final paper, and release the code as promised.